# A Selective Learning Method for Temporal Graph Continual Learning

Hanmo LIU [1 2]  Shimin DI [3]  Haoyang LI [4]  Xun JIAN [5]  Yue WANG [6]  Lei CHEN [2 1]

## Abstract

Node classification is a key task in temporal graph learning (TGL). Real-life temporal graphs often introduce new node classes over time, but existing TGL methods assume a fixed set of classes. This assumption brings limitations, as updating models with full data is costly, while focusing only on new classes results in forgetting old ones. Graph continual learning (GCL) methods mitigate forgetting using old-class subsets but fail to account for their evolution. We define this novel problem as temporal graph continual learning (TGCL), which focuses on efficiently maintaining up-to-date knowledge of old classes. To tackle TGCL, we propose a selective learning framework that substitutes the old-class data with its subsets, Learning Towards the Future (LTF). We derive an upper bound on the error caused by such replacement and transform it into objectives for selecting and learning subsets that minimize classification error while preserving the distribution of the full old-class data. Experiments on three real-world datasets validate the effectiveness of LTF on TGCL.

## 1. Introduction

Temporal graphs are essential data structures for real-world applications, such as social networks (Baumgartner et al., 2020) and online shopping (Ni et al., 2019). In temporal graphs, the edges and/or nodes change over time, with these additions or deletions captured as a sequence of events (Yang et al., 2023; de Barros et al., 2023). In recent years, various temporal graph learning (TGL) methods have been developed to extract insights from temporal graphs by incorporating temporal-neighbor information into node em-

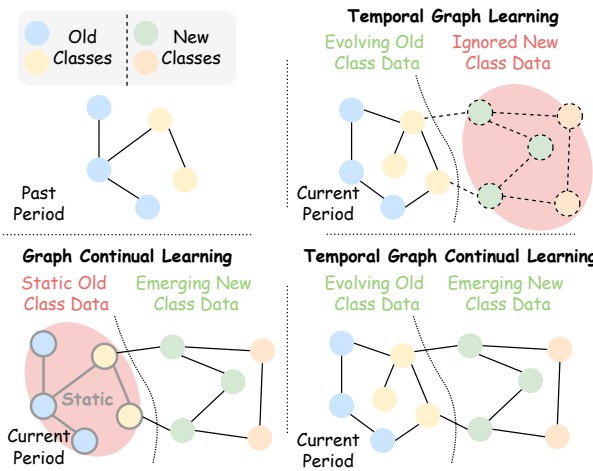

Figure 1: The differences in temporal graph learning (TGL), graph continual learning (GCL) and temporal graph continual learning (TGCL). At a new period, TGL assumes no data of new classes appear, while GCL assumes static old-class data. TGCL holds neither of these assumptions, thus is more suitable to real-life temporal graphs.

beddings (Kumar et al., 2019; Rossi et al., 2020; Xu et al., 2020; Cong et al., 2023; Wen & Fang, 2022; Li & Chen, 2023; Li et al., 2023). The current approaches for processing both structural and temporal information in these graphs utilize a range of model architectures, including message-passing mechanisms (Rossi et al., 2020; Xu et al., 2020; Wen & Fang, 2022), multi-layer perceptrons (MLPs) (Cong et al., 2023; Gardner & Dorling, 1998), and transformers (Yu et al., 2023; Vaswani et al., 2017). A key application of TGL methods is node classification, which is a critical task in the analysis of temporal graphs (Rossi et al., 2020; Xu et al., 2020; Yu et al., 2023). For example, in social networks, TGL methods classify normal and malicious users based on their interactions.

While TGL methods are effective at classifying nodes, they face a significant limitation: they assume a static set of node classes, which does not reflect the dynamic reality of these environments. This static assumption is illustrated in Fig. 1, where node classes under the TGL setting remain unchanged over time. However, real-world scenarios often exhibit an open class setting, where new classes frequently emerge. Like in social networks, new malicious behaviours contin-

---

[1]Hong Kong University of Science and Technology, China [2]Hong Kong University of Science and Technology (Guangzhou), China [3]Southeast University, China [4]Hong Kong Polytechnic University, China [5]Northwestern Polytechnical University, China [6]Shenzhen Institute of Computing Sciences, China. Correspondence to: Shimin DI <shimin.di@seu.edu.cn>.

*Proceedings of the 42^{nd} International Conference on Machine Learning*, Vancouver, Canada. PMLR 267, 2025. Copyright 2025 by the author(s).

ually arise, introducing new node classes into the system (Feng et al., 2023). Due to the fixed class set assumption, adapting current TGL methods to this open class setting presents efficiency and effectiveness challenges. Updating the model for all classes becomes inefficient as the temporal graph grows, while fine-tuning the model for only new classes risks catastrophic forgetting (French & Chater, 2002; Parisi et al., 2019; Masana et al., 2023) of older classes, particularly when their data distribution diverges from past instances.

To address the issue of forgetting when fine-tuning TGL models, continual learning (Parisi et al., 2019; Masana et al., 2023) provides a promising solution. Recently, several graph continual learning (GCL) methods have been proposed to preserve old-class knowledge by either regularizing model parameters associated with previous classes (Liu et al., 2021) or replaying representative subsets of old-class data (Kim et al., 2022; Zhou & Cao, 2021; Chen et al., 2021; Wang et al., 2022; Zhang et al., 2022; Feng et al., 2023). However, existing GCL methods struggle to handle open-class temporal graphs. The major limitation is that they assume the seen data will be static in the future, as shown in Fig.1. Such an assumption contradicts with the dynamic nature of temporal graphs, making the model become out-dated for future temporal graphs.

These limitations in current TGL and GCL methods make updating models for open-class temporal graphs a challenging problem, which we define as temporal graph continual learning (TGCL). The **challenge** in TGCL is how to maintain both the effectiveness and recency of old-class knowledge while ensuring high efficiency. To address this, we propose a selective learning method that identifies and learns representative subsets of old-class data in new temporal graphs, named *L*earning *T*owards the *F*uture (LTF). While subset selection is a common approach, detailed analysis of how well these subsets represent the full dataset remains limited, especially when learning from the entire dataset is impractical. To address this, we derive an upper bound on the error introduced by approximating the full dataset with a subset. We transform the upper bound into a subset selection objective to minimize this error. Additionally, guided by the upper bound, we design a regularization loss that aligns the embedding distribution of the selected subset with that of the full dataset to pursue better performance. Our contribution can be summarized as follows:

- We are among the first to investigate how to effectively and efficiently update a model on temporal graphs with emerging new-class data and evolving old-class data, which we term the temporal graph continual learning (TGCL) problem.
- Selecting representative old-class subsets is crucial for addressing the TGCL problem. To achieve this, we define

a selection objective that minimizes the upper-bound error on the old classes.

- The knowledge from the subsets is hard to generalize to the full old-class data. We solve this problem by designing an efficient loss that aligns the distributions of the subset and the full data.

- We conduct extensive experiments on real-world web data, Yelp, Reddit, and Amazon. The results show that our method is effective while ensuring high efficiency.

## 2. Background

Many real-world scenarios are modeled as temporal graphs (Yang et al., 2023; de Barros et al., 2023; Kazemi et al., 2020), such as social networks and online shopping networks. In this paper, the temporal graph $G = (V, E, T, Y) \sim \mathcal{G}$ is a set of nodes $V$ with labels $Y$ connected by time-stamped events $E$ happening among $V$ within the time period $T$. $G$ follows the distribution $\mathcal{G}$. Each event $e = \{u_t, v_t, t\} \in E$ is an interaction (edge) between two nodes $u_t, v_t \in V$ at time $t \in T$. $G$ can be equivalently expressed as $E$. Each node $v_t \in V$ is related with time $t$ and has a time-dependent feature $\mathbf{x}_t$.

Suppose the temporal graph has evolved for $N$ time periods $\{T_1, T_2, ..., T_N\}$, and the corresponding temporal graphs are noted as $\{G_1, G_2, ..., G_N\}$ which follow the distributions $\{\mathcal{G}_1, \mathcal{G}_2, ..., \mathcal{G}_N\}$. Each period has a new set of classes $\{Y_1, Y_2, ..., Y_N\}$, where $Y_i \cap Y_j = \emptyset, \forall i, j \leq N$ and $i \neq j$. For simplicity, we note the old classes at $T_N$ as $Y_{old} = \cup\{Y_n\}_{n<N}$. Corresponding to the node classes, $G_N$ can be separated into $G_N^{new} = (V_N^{new}, E_N^{new}, T_N, Y_N)$ and $G_N^{old} = (V_N^{old}, E_N^{old}, T_N, Y_{old})$, where $G_N = G_N^{old} \cup G_N^{new}$, $V_N^{old} \cap V_N^{new} = \emptyset$ and $E_N^{old} \cap E_N^{new} \neq \emptyset$. $E_N^{old}$ and $E_N^{new}$ are overlapping at the events connecting between $V_N^{old}$ and $V_N^{new}$. It is worth noting that $G_N^{old}$ has the same set of classes as $G_{N-1}$, but the data distribution is different due to the temporal evolution. The illustration on relationships among $G_{N-1}$, $G_N^{old}$ and $G_N^{new}$ and the notation summary are presented in Appendix A.

### 2.1. Temporal Graph Learning

In recent years, many temporal graph learning methods are proposed (Parisi et al., 2019; Masana et al., 2023) to extract knowledge from the temporal graphs. Under the fixed class set assumption, at a new period $T_N$, the TGL methods under the node classification task aims to minimize the classification error of the model $h$ on $G_N^{old}$, i.e. the part of $G_N$ with only $Y_{old}$. Suppose that $h$ is a binary classification hypothesis (model) from the hypothesis space $\mathcal{H}$ with finite VC-dimension, the TGL objective is formulated as:

$$\tilde{h}_N = \arg\min_{h \in \mathcal{H}} \epsilon(h | \mathcal{G}_N^{old}), \tag{1}$$

where $\epsilon(h|\mathcal{G}) := \mathbb{E}_{v_t \in \mathcal{G}}[h(v_t) \neq f(v_t)]$ is the expected classification error of $h$ on any distribution $\mathcal{G}$, and $f(\cdot)$ is an unknown deterministic function that gives the ground truth classification on each $v_t$.

Early TGL works (Wu et al., 2021; Skarding et al., 2021) integrate events of temporal graphs into a sequence of snapshots, which loses fine-grained continuous-time information. Thus recent TGL methods preserve events as the basic training instances (Yang et al., 2023). As the pioneers, JODIE (Kumar et al., 2019) processes and updates the embeddings of each node based on their related events by using a recursive neural network (Alom et al., 2019). CTDNE (Nguyen et al., 2018) and CAW (Wang et al., 2021a) aggregate the information through random walks on the events. TGAT (Rossi et al., 2020), TGN (Xu et al., 2020) and TREND (Wen & Fang, 2022) apply the GNN-like message-passing mechanism to capture the temporal and structural information at the same time. More recently, there are also works using the multi-layer perceptrons (Cong et al., 2023) or transformers (Yu et al., 2023) to understand the temporal graphs. Other than structure designs, temporal learning techniques like the temporal point process (Trivedi et al., 2019; Wen & Fang, 2022) are also integrated into the TGL methods to better capture the temporal dynamics.

## 2.2. Graph Continual Learning

As new classes continuously emerge for real-life temporal graphs, how to efficiently learn new classes without forgetting the old knowledge (French & Chater, 2002) becomes an important problem. For Euclidean data like images, the forgetting issue has been addressed by many continual learning methods (Parisi et al., 2019; Masana et al., 2023). The common approaches include regularizing the model parameters , replaying the subsets of the old data , or adjusting the model parameters . Recently, some GCL methods are trying to connect continual learning to dynamic graphs (Tian et al., 2024). The objective of the GCL problem is:

$$\tilde{h}_N = \arg \min_{h \in \mathcal{H}} \epsilon(h|\mathcal{G}_N^{new}) + \epsilon(h|\mathcal{G}_{N-1}), \qquad (2)$$

where $\mathcal{G}_N^{new}$ and $\mathcal{G}_{N-1}$ are distributions of $G_N^{new}$ and $G_{N-1}$, and the former term is for learning new-class knowledge while the latter one is for maintaining old-class knowledge. To reduce the cost of learning $G_{N-1}$, most of the GCL methods try to use subsets of $G_{N-1}$ to approximate its error, where the subsets are selected by the node influence (Zhou & Cao, 2021) or the structural dependency (Chen et al., 2021; Kim et al., 2022; Zhang et al., 2022), or generated via auxiliary models (Wang et al., 2022). TWP (Liu et al., 2021) takes a different approach by preventing the important parameters for classification and message-passing from being modified. SSRM (Su et al., 2023a) aligns distributions between old and new class data distributions for better

performances. However, these methods primarily target graph snapshots and are less effective for event-based temporal graphs (Feng et al., 2023). This gap is first addressed by OTGNet (Feng et al., 2023), which proposes to replay important and diverse triads (Zhou et al., 2018) and learn class-agnostic embeddings.

## 3. Methodology

TGCL problem takes a more realistic temporal graph setting that considers both the appearance of new classes and the evolution of old-class data. At a new period $T_N$, TGCL requires the model $h$ to learn the new classes from $G_N^{new}$ and maintain the old-class knowledge from $G_N^{old}$:

$$\tilde{h}_N = \arg\min_{h \in \mathcal{H}} \epsilon(h|\mathcal{G}_N) = \arg\min_{h \in \mathcal{H}} \epsilon(h|\mathcal{G}_N^{new}) + \epsilon(h|\mathcal{G}_N^{old}). \quad (3)$$

Compared with the TGL problem at Eq. (1), the TGCL problem additionally minimizes $\epsilon(h|\mathcal{G}_N^{new})$. Besides, TGCL maintains more recent old-class knowledge from $G_N^{old}$, differing from $G_{N-1}$ in the GCL problem at Eq. (2).

In this work, we focus on how to achieve both effectiveness and efficiency in minimizing $\epsilon(h|\mathcal{G}_N^{old})$, and directly minimize $\epsilon(h|\mathcal{G}_N^{new})$ as most continual learning works do. We follow a common strategy by selecting and learning a subset $G_N^{sub}$ of $G_N^{old}$. To obtain an optimal performance, we first derive an upper bound on the error introduced by approximating $G_N^{old}$ with $G_N^{sub}$ in Sec.3.1. We then transform this theoretical bound into a tractable optimization problem to facilitate subset selection in Sec.3.2. Lastly, this error is further minimized during learning by aligning the distribution of $G_N^{sub}$ with $G_N^{old}$, as detailed in Sec.3.3. The framework of LTF is illustrated in Fig.2.

### 3.1. Classification Error Upper-bound

A small classification error $\epsilon(h|\mathcal{G}_N^{old})$ on $G_N^{old}$ is essential for the model effectiveness. Selecting and learning $G_N^{sub} \subset G_N^{old}$ assumes that minimizing $\epsilon(h|\mathcal{G}_N^{sub})$ will also minimize $\epsilon(h|\mathcal{G}_N^{old})$. While heuristics can help align these errors, a theoretical analysis connecting them is lacking. We address this gap using domain adaptation theory (Redko et al., 2020a; Ben-David et al., 2010) and show that $\epsilon(h|\mathcal{G}_N^{sub})$ can approximate $\epsilon(h|\mathcal{G}_N^{old})$ with upper-bounded additional error.

Based on Lemma 3 from (Ben-David et al., 2010) (see Appendix C), the classification disagreement of any two models $h$ and $h'$ on any two data distributions is upper-bounded by the discrepancy between those two distributions. Then, the upper-bound on the classification error of $\mathcal{G}_N^{old}$ can be derived as the following theorem:

**Theorem 3.1.** *Let $\mathcal{G}_N^{old}, \mathcal{G}_N^{sub}$ be the distributions of $G_N^{old}$ and $G_N^{sub}$. Let $h \in \mathcal{H}$ be a function in the hypothesis space*

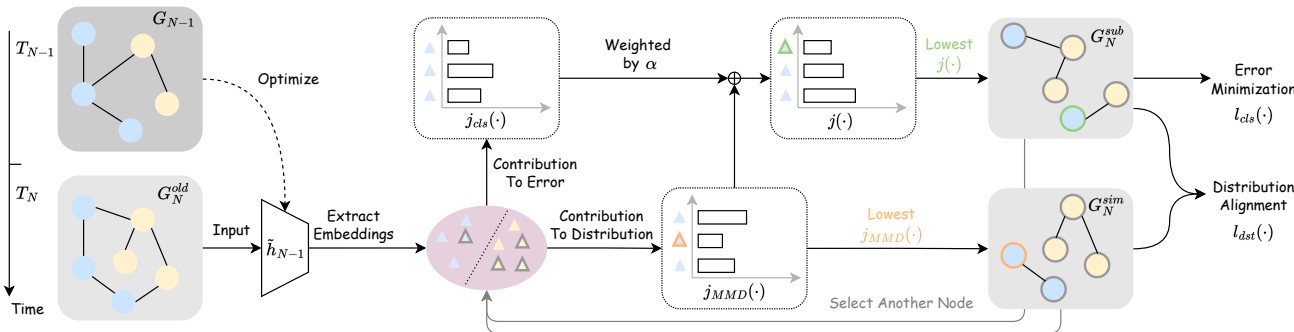

Figure 2: The selective learning framework of LTF on old-class data. From $G_N^{old}$, $G_N^{sub}$ is greedily selected by having the lowest classification error $j_{cls}(\cdot)$ and distribution discrepancy $j_{MMD}(\cdot)$, while $G_N^{sim}$ is greedily selected only by the lowest $j_{MMD}(\cdot)$. Afterwards, $G_N^{sub}$ is learned by minimizing the classification error and aligning the distribution with $G_N^{sim}$.

$\mathcal{H}$ and $\tilde{h}_N^{sub}$ be the function optimized on $\mathcal{G}_N^{sub}$. The classification error on $\mathcal{G}_N^{old}$ then has the following upper bound:

$$\min_{h \in \mathcal{H}} \epsilon(h | \mathcal{G}_N^{old}) \leq \min_{h, \mathcal{G}_N^{sub}} \epsilon(\tilde{h}_N^{sub} | \mathcal{G}_N^{old}) \tag{4}$$

$$+ \frac{1}{2} d_{\mathcal{H}\Delta\mathcal{H}}(\mathcal{G}_N^{old}, \mathcal{G}_N^{sub}) + \epsilon(h, \tilde{h}_N^{sub} | \mathcal{G}_N^{sub}),$$

where $d_{\mathcal{H}\Delta\mathcal{H}}(\mathcal{G}_a, \mathcal{G}_b) = 2 \sup_{h \in \mathcal{H}\Delta\mathcal{H}} |Pr_{v_t \sim \mathcal{G}_a}[h(v_t) = 1] - Pr_{v_t \sim \mathcal{G}_b}[h(v_t) = 1]|$ *measures the $\mathcal{H}\Delta\mathcal{H}$ divergence between the distributions $\mathcal{G}_a$ and $\mathcal{G}_b$, and $\epsilon(h, h'|\mathcal{D}) := \mathbb{E}_{x \in \mathcal{D}}[h(x) \neq h'(x)]$ is the expected prediction differences of $h$ and $h'$ on $\mathcal{D}$.*

The proof is given in Appendix C. Following Theorem 3.1, there are three criteria to ensure that $h$ achieves a lower error on $G_N^{old}$ by finding a suitable $G_N^{sub}$:

- Small error $\epsilon(\tilde{h}_N^{sub} | \mathcal{G}_N^{old})$ indicates that $\tilde{h}_N^{sub}$ learned on subset $G_N^{sub}$ is also predictive on the entire old data $G_N^{old}$;
- Small distribution difference $d_{\mathcal{H}\Delta\mathcal{H}}(\mathcal{G}_N^{old}, \mathcal{G}_N^{sub})$ indicates that the subset $G_N^{sub}$ is diverse enough to represent the entire old class data $G_N^{old}$;
- Small error $\epsilon(h, \tilde{h}_N^{sub} | \mathcal{G}_N^{sub})$ indicates that $h$ classifies $\mathcal{G}_N^{sub}$ similarly as $\tilde{h}_N^{sub}$, which guarantees that the knowledge of $\tilde{h}_N^{sub}$ can be transferred to $h$.

By relaxing $\epsilon(h | \mathcal{G}_N^{old})$ to the upper bound in Eq. (4), we convert the original problem into selecting subset $G_N^{sub}$ (Sec. 3.2) and updating the TGCL model $h$ (Sec. 3.3).

### 3.2. Subset Selection

Theorem 3.1 specifies three criteria for selecting $G_N^{sub}$ from $G_N^{old}$. Since $h$ is unknown during selection, third criterion is omited in following analysis. The remaining two criteria for a representative $G_N^{sub}$ are: 1) a low $\epsilon(\tilde{h}_N^{sub} | \mathcal{G}_N^{old})$, reflecting strong predictive performance on $G_N^{old}$, and 2) a low $d_{\mathcal{H}\Delta\mathcal{H}}(\mathcal{G}_N^{old}, \mathcal{G}_N^{sub})$, ensuring close alignment with the distribution of $G_N^{old}$. Next, we introduce how we transform these two criteria into a tractable optimization problem for selecting representative $G_N^{sub}$.

**Base Algorithm.** In minimizing $\epsilon(\tilde{h}_N^{sub} | \mathcal{G}_N^{old})$, it is infeasible to obtain $\tilde{h}_N^{sub}$ from each possible $\mathcal{G}_N^{sub}$ for selection. Therefore we approximate $\tilde{h}_N^{sub}$ with $\tilde{h}_{N-1}$ that is optimized on $\mathcal{G}_{N-1}$. This is a reasonable approximation, as $\tilde{h}_{N-1}$ is readily available and $G_{N-1}$ closely resembles the full data $G_N^{old}$ by sharing the same class set $Y_{old}$ and being temporally proximate. With $\tilde{h}_N^{sub}$ substitued by $\tilde{h}_{N-1}$, based on the inequality in (Ben-David et al., 2010), $\epsilon(\tilde{h}_N^{sub} | \mathcal{G}_N^{old})$ is expanded to: $\epsilon(\tilde{h}_N^{sub} | \mathcal{G}_N^{old}) \leq \epsilon(\tilde{h}_N^{sub}, \tilde{h}_{N-1} | \mathcal{G}_N^{old}) + \epsilon(\tilde{h}_{N-1} | \mathcal{G}_N^{old})$. As $\epsilon(\tilde{h}_{N-1} | \mathcal{G}_N^{old})$ is unrelated to the subset, we focus on minimizing $\epsilon(\tilde{h}_N^{sub}, \tilde{h}_{N-1} | \mathcal{G}_N^{old})$, which requires aligning $\tilde{h}_N^{sub}$ with $\tilde{h}_{N-1}$. Because $\tilde{h}_N^{sub}$ depends on $G_N^{sub}$, we select $G_N^{sub}$ that minimizes $\epsilon(\tilde{h}_{N-1} | \mathcal{G}_N^{sub})$ to match the behaviors of $\tilde{h}_N^{sub}$ and $\tilde{h}_{N-1}$. This simplifies our selection problem to:

$$\tilde{\mathcal{G}}_N^{sub} = \arg \min_{\mathcal{G}_N^{sub}} \epsilon(\tilde{h}_{N-1} | \mathcal{G}_N^{sub}) + d_{\mathcal{H}\Delta\mathcal{H}}(\mathcal{G}_N^{old}, \mathcal{G}_N^{sub}). \tag{5}$$

Due to limited available observations in reality, we transform the distribution-level error $\epsilon(\tilde{h}_{N-1} | \mathcal{G}_N^{sub})$ into the empirical error $\hat{\epsilon}(\cdot)$ on the finite subest $G_N^{sub}$:

$$\hat{\epsilon}(\tilde{h}_{N-1} | G_N^{sub}) = \frac{1}{|G_N^{sub}|} \sum_{(v_t, y) \in G_N^{sub}} l_{cls}(\tilde{h}_{N-1}(v_t), y), \tag{6}$$

where $l_{cls}(\cdot)$ is the classification error like the mean square error. Similarly, we estimate $d_{\mathcal{H}\Delta\mathcal{H}}(\mathcal{G}_N^{old}, \mathcal{G}_N^{sub})$ by the square of the mean maximum distribution (MMD) (Gretton et al., 2006) on the finite sets $G_N^{old}$ and $G_N^{sub}$:

$$\hat{d}_{MMD}^2(G_a, G_b) = \frac{1}{|G_a|^2} \sum_{v_t, u_t \in G_a} k(v_t, u_t) \tag{7}$$

$$- \frac{2}{|G_a||G_b|} \sum_{v_t \in G_a, u_t \in G_b} k(v_t, u_t) + \frac{1}{|G_b|^2} \sum_{v_t, u_t \in G_b} k(v_t, u_t),$$

where $k(\cdot, \cdot)$ is the kernel function, $G_a$ and $G_b$ are adopted for simplifying notations. To evaluate the kernel values, we

take a common practice to replace the raw data by their embeddings (Su et al., 2023b; Shi & Wang, 2023; Redko et al., 2020b), which are extracted by $\tilde{h}_{N-1}$ and noted as $\hat{d}^2_{MMD}(G^{sub}_N, G^{old}_N | \tilde{h}_{N-1})$.

With above estimations, the selection objective of $G^{sub}_N$ in Eq.(5) is transformed into:

$$\tilde{G}^{sub}_N = \arg \min_{|G^{sub}_N| \leq m, G^{sub}_N \subset G^{old}_N} \alpha \hat{\epsilon}(\tilde{h}_{N-1} | G^{sub}_N) \quad (8)$$
$$+ \hat{d}^2_{MMD}(G^{old}_N, G^{sub}_N | \tilde{h}_{N-1}),$$

where $\alpha$ is a weight hyper-parameter and $m$ is the memory budget for $G^{sub}_N$. A larger $m$ brings a better estimation but also increases the computation complexity.

**Greedy Algorithm.** Directly optimizing the selection objective in Eq. (8) is infeasible due to its factorial time complexity. However, it can be proven to be a monotone submodular function, allowing greedy optimization with a guaranteed approximation to the optimal solution.

The first term $\hat{\epsilon}(\tilde{h}_{N-1} | G^{sub}_N)$ is linear to the classification error of each instance, thus it is directly monotone submodular. Following the proof in (Kim et al., 2016), $\hat{d}^2_{MMD}(G^{old}_N, G^{sub}_N | \tilde{h}_{N-1})$ is also monotone submodular function with respect to $G^{sub}_N$, provided $k(v_t, u_t)$ satisfies $0 \leq k(v_t, u_t) \leq k(v_t, v_t)/(|G^{old}_N|^3 - 2|G^{old}_N|^2 - 2|G^{old}_N| - 3)$, $\forall v_t, u_t \in G^{old}_N$, $v_t \neq u_t$. This requirement is met with a properly parameterized kernel, and we use the Radial Basis Function kernel (Schölkopf et al., 1997) in practice. Thus, Eq. (8), as a sum of two monotone submodular functions, is itself monotone submodular based on the theory in (Cook et al., 2011). Consequently, as per (Nemhauser et al., 1978), Eq.(8) can be efficiently approximated by a greedy algorithm, achieving an error bound of $(1 - 1/e)$ relative to the optimal solution.

To implement the greedy algorithm, we derive the witness function $j(\cdot)$ to evaluate how adding one node to $G^{sub}_N$ affects the value of Eq.(8). The function $j(\cdot)$ is a summation of two separate witness functions $j_{cls}(\cdot)$ and $j_{MMD}(\cdot)$, corresponding to $\hat{\epsilon}(\tilde{h}_{N-1} | G^{sub}_N)$ and $\hat{d}^2_{MMD}(G^{old}_N, G^{sub}_N)$. Since the classification error is a summation over node-wise losses defined in Eq.(6), $j_{cls}(v_t) = l_{cls}(v_t, y | \tilde{h}_{N-1})$. On the other hand, $j_{MMD}(\cdot)$ can be derived from Eq. (7) as:

$$j_{MMD}(v_t) = \frac{2}{|G^{sub}_N|} \sum_{u_t \in G^{sub}_N} k(v_t, u_t | \tilde{h}_{N-1}) \quad (9)$$
$$- \frac{2}{|G^{old}_N|} \sum_{u_t \in G^{old}_N} k(v_t, u_t | \tilde{h}_{N-1}),$$

where $k(v_t, u_t | h)$ notes the kernel calculated by the node embeddings extracted from the model $h$. Then, the overall

witness function $j(\cdot)$ is expressed as:

$$j(v_t) = \alpha j_{cls}(v_t) + j_{MMD}(v_t),$$

where $\alpha$ is the same as in Eq. (8). Afterwards, we greedily select the nodes with the smallest $j(\cdot)$ from $G^{old}_N$ to $G^{sub}_N$ untill the buffer is full.

**Cost Reduction.** During greedy selection, estimating $\hat{d}^2_{MMD}(G^{old}_N, G^{sub}_N | \tilde{h}_{N-1})$ has a high computational complexity of $O((|G^{old}_N| + |G^{sub}_N|)^2)$, which limits its application to large data sets. Thus, we propose to evenly partition $G^{old}_N$ into groups of size $p$, $|G^{old}_N| > p \gg m$, resulting in $W = \lceil |G^{old}_N|/p \rceil$ partitions. We then greedily select $1/W$ of $G^{sub}_N$ from each partition and join them as the final subset. The complexity of selecting $G^{sub}_N$ from each partition is reduced to $O((|G^{old}_N| + |G^{sub}_N|)^2/W^2)$.

Based on the triangle inequality of $d_{\mathcal{H} \Delta \mathcal{H}}(\cdot, \cdot)$ (Gretton et al., 2006), this partition procedure enlarges the second term of Eq. (8) to $d_{\mathcal{H} \Delta \mathcal{H}}(\mathcal{G}^{old}_N, \mathcal{G}^{sub}_N) \leq d_{\mathcal{H} \Delta \mathcal{H}}(\mathcal{G}^{old}_N, \mathcal{G}^{old}_{N,w}) + d_{\mathcal{H} \Delta \mathcal{H}}(\mathcal{G}^{old}_{N,w}, \mathcal{G}^{sub}_{N,w})$ for each partition $G^{old}_{N,w} \sim \mathcal{G}^{old}_{N,w}$. To reduce this additional error, $\mathcal{G}^{old}_{N,w}$ should be similar to $\mathcal{G}^{old}_N$. As the partitioned data can remain a large size for the subset selection, the random partition can well satisfy this requirement.

### 3.3. Model Optimization

After selecting an optimal subset from Sec. 3.2, we transform Eq. (4) into a concrete learning objective to train an effective model $h$ with $G^{sub}_N$. Because $\tilde{h}^{sub}_N$ is determined after the subset is selected and $\mathcal{G}^{old}_N$ is fixed, the first term $\epsilon(\tilde{h}^{sub}_N | \mathcal{G}^{old}_N)$ of Eq. (4) cannot be further optimized and is omitted. We minimize $d_{\mathcal{H} \Delta \mathcal{H}}(\mathcal{G}^{old}_N, \mathcal{G}^{sub}_N)$ by enclosing the embedding distributions of both data sets extracted by $h$ (Ben-David et al., 2010; Su et al., 2023b; Shi & Wang, 2023; Redko et al., 2020b), i.e., minimize $d_{\mathcal{H} \Delta \mathcal{H}}(\mathcal{G}^{old}_N, \mathcal{G}^{sub}_N | h)$. By replacing the population terms with their estimations, the objective of learning $G^{sub}_N$ is $\hat{\epsilon}(h, \tilde{h}^{sub}_N | G^{sub}_N) + d_{MMD}(G^{old}_N, G^{sub}_N | h)$.

As we avoid learning $G^{old}_N$ for efficiency reason, we substitute $G^{old}_N$ with its similarly distributed subset $G^{sim}_N \subset G^{old}_N$, $\mathcal{G}^{sim}_N \approx \mathcal{G}^{old}_N$. To satisfy this requirement, $G^{sim}_N$ is selected by solely minimizing its distribution discrepancy with $G^{old}_N$:

$$\tilde{G}^{sim}_N = \arg \min_{\substack{|G^{sim}_N| \leq m' \\ G^{sim}_N \subset G^{old}_N}} \hat{d}^2_{MMD}(G^{sim}_N, G^{old}_N | \tilde{h}_{N-1}), \quad (10)$$

where $m'$ is the memory budget for $G^{sim}_N$. A larger $m'$ improves the estimation but also increases the computation complexity. $G^{sim}_N$ can be selected by greedily finding the nodes with the lowest $j_{MMD}(\cdot)$ defined in Eq. (9). We further reduce the selection cost of $G^{sim}_N$ by data partitioning,

Table 1: Data statistics

| Data Set | Yelp | Reddit | Amazon |
|---|---|---|---|
| # Nodes | 19,918 | 13,106 | 84,605 |
| # Events | 2,321,186 | 310,231 | 875,563 |
| # Timespan / Period | 1 year | 20 days | 24 days |
| # Periods | 5 | 3 | 3 |
| # Classes / Period | 3 | 5 | 3 |
| # Total Classes | 15 | 15 | 9 |

similar to $G_N^{sub}$. After substituting $G_N^{old}$ with $G_N^{sim}$, the learning objective is transformed to:

$$l_{old}(G_N^{sim}, G_N^{sub}|h) = \hat{\epsilon}(h|G_N^{sub}) + \hat{d}_{MMD}(G_N^{sim}, G_N^{sub}|h), \quad (11)$$

where $\hat{\epsilon}(h, \tilde{h}_N^{sub}|G_N^{sub})$ is written as $\hat{\epsilon}(h|G_N^{sub})$ since both terms are equivalent in making $h$ perform like $\tilde{h}_N^{sub}$.

In practice, the complexity of calculating $\hat{d}_{MMD}^2(G_N^{sim}, G_N^{sub}|h)$ is $O((|G_N^{sim}| + |G_N^{sub}|)^2)$, which is much higher than $O(|G_N^{sub}|)$ of the classification error calculation. So that we further simplify its format to improve the efficiency. To ensure that $G_N^{sim}$ is the target distribution of $G_N^{sub}$ during optimization, we stop the gradients of $G_N^{sim}$ from being back-propagated. Following this stop gradient design, the first and third terms in $\hat{d}_{MMD}^2(\cdot)$ definition at Eq. (7) are omitted, since $G_N^{sim}$ is not learned and the self-comparison within $G_N^{sub}$ is meaningless. After this simplification, the complexity is reduced from $O((|G_N^{sim}| + |G_N^{sub}|)^2)$ to $O(|G_N^{sim}||G_N^{sub}|)$, and $\hat{d}_{MMD}^2(G_N^{sim}, G_N^{sub}|h)$ is optimized by:

$$l_{dst}(G_N^{sub}, G_N^{sim}|h) = -s \sum_{\substack{v_t \in G_N^{sub} \\ u_t \in G_N^{sim}}} k(v_t, \text{sg}(u_t)|h),$$

where $s = 2/|G_N^{sub}| \cdot |G_N^{sim}|$ and $\text{sg}(\cdot)$ means stopping the gradients from back-propagation.

Finally, by including the objective of learning $G_N^{new}$, our total objective of updating $h$ at $T_N$ is:

$$l_{tot} = \hat{\epsilon}(h|G_N^{new}) + \hat{\epsilon}(h|G_N^{sub}) + \beta l_{dst}(G_N^{sim}, G_N^{sub}|h),$$

where $\beta$ is the hyper-parameter weighting the distribution regularization importance. The pseudo-code of our overall framework is presented in Algorithm 1 at Appendix D.

# 4. Experiments

## 4.1. Experiment Setup

**Data Set.** [1] We evaluate our method using three real-world datasets: Yelp (dat), Reddit (Baumgartner et al., 2020), and

---

[1]Our code and data are available at https://github.com/liuhanmo321/TGCL_LTF.git.

Amazon (Ni et al., 2019). Yelp, a business-to-business temporal graph from 2014 to 2019, treats businesses in the same category as nodes of the same class, with user interactions creating events. The graph is divided into five periods, each representing a year, with three new business categories introduced each year. Reddit, a post-to-post temporal graph, treats subreddit topics as classes and posts as nodes. User comments create events, with every 24 days forming a period and five new subreddits introduced each period. Amazon is constructed similarly to Yelp, with 20-day periods and three new businesses per period. The temporal graph transformation mechanism is similar to OTGNet (Feng et al., 2023), but adapted to our unique problem definition. Dataset statistics are summarized in Tab. 1, with additional details in the Appendix G.

**Backbone Model.** As LTF is agnostic to TGL model designs, we select the classic model TGAT (Rossi et al., 2020) and the state-of-the-art model DyGFormer (Yu et al., 2023) as our backbone models. TGAT uses the self-attention mechanism to aggregate the temporal neighbor information and embed the nodes. DyGFormer applies the transformer structure and uses the structural encoding to embed the nodes.

**Baselines.** First, we select three classic continual learning models that are adaptable to the TGCL problem, which are EWC (Kirkpatrick et al., 2016), LwF (Li & Hoiem, 2018) and iCaRL (Rebuffi et al., 2017). EWC and LwF use regularization losses to prevent forgetting the old class knowledge while not using the old class data. iCaRL selects the representative old class data based on the closeness to the mean of the embeddings. For the GCL methods, we select the replay-based methods ER (Zhou & Cao, 2021), SSM (Zhang et al., 2022), OTGNet (Feng et al., 2023), and URCL (Miao et al., 2024). Besides, the naive baselines of learning the full $G_N$ (Joint) and learning only the $G_N^{new}$ (Finetune) are included. Joint is the upper-bound for performance with the lowest efficiency, while Finetune is the opposite.

**Evaluation Metric.** The average precision (AP) and average forgetting (AF) on each set of classes within a period are used to evaluate the model performance. In $G_n$, there are $n$ sets of classes $\{Y_1, ..., Y_n\}$, and the model's precision on each of them is $P_{n,i}, \forall i \leq n$. Then AP at period $T_n$ is calculated as $AP_n := \frac{1}{n} \sum_{i=1}^{n} P_{n,i}$. To evaluate the forgetting issue at $T_n$, we use the precision difference between the current method and Joint ($P_{n,i}^{jnt}$) as the forgetting score for $Y_i$, which is $F_{n,i} := P_{n,i}^{jnt} - P_{n,i}$. Then the AF at period $T_n$ is calculated as $AF_n := \frac{1}{n-1} \sum_{i=1}^{n-1} F_{n,i}$. For simplicity, we omit the subscript $N$ for $AP_N$ and $AF_N$ as they reflect the final performance. A higher value of AP is better, while a lower value of AF is better. To evaluate the efficiency, the average training time per epoch (abbreviated as *Time*) at the

Table 2: The comparison between LTF and the baselines methods. The best and second best results are noted in **Bold** and Underline. Joint and Finetune are excluded from the notations.

| Method | TGAT | | | | | | | | | DyGFormer | | | | | | | | |
| --- | --- | --- | --- | --- | --- | --- | --- | --- | --- | --- | --- | --- | --- | --- | --- | --- | --- | --- |
| | Yelp | | | Reddit | | | Amazon | | | Yelp | | | Reddit | | | Amazon | | |
| | AP↑ | AF↓ | Time↓ | AP↑ | AF↓ | Time↓ | AP↑ | AF↓ | Time↓ | AP↑ | AF↓ | Time↓ | AP↑ | AF↓ | Time↓ | AP↑ | AF↓ | Time↓ |
| Joint | 0.0810 | — | 58.37 | 0.1378 | — | 50.50 | 0.1477 | — | 128.71 | 0.0813 | — | 95.11 | 0.1256 | — | 70.64 | 0.1500 | — | 177.38 |
| Finetune | 0.0141 | 0.0843 | 9.11 | 0.0312 | 0.1550 | 14.93 | 0.0340 | 0.1408 | 65.81 | 0.0172 | 0.0800 | 14.43 | 0.0360 | 0.1433 | 20.58 | 0.0551 | 0.1517 | 88.34 |
| LwF | 0.0209 | 0.0620 | 13.90 | 0.0439 | 0.1091 | 23.67 | 0.0303 | 0.1024 | 102.92 | 0.0399 | 0.0386 | 26.03 | 0.0469 | 0.0944 | 37.53 | 0.0763 | 0.0856 | 155.03 |
| EWC | 0.0443 | 0.0408 | **9.19** | 0.0467 | 0.1384 | **14.95** | 0.0524 | 0.1152 | 68.37 | 0.0601 | 0.0295 | **14.24** | 0.0521 | 0.1046 | **20.20** | 0.1005 | 0.0832 | **89.32** |
| iCaRL | 0.0607 | 0.0198 | 11.57 | 0.0602 | 0.0860 | 19.22 | 0.0699 | 0.0794 | 70.03 | 0.0558 | 0.0214 | 18.31 | 0.0917 | 0.0248 | 26.34 | 0.0945 | 0.0775 | 92.36 |
| ER | 0.0521 | 0.0332 | 11.63 | 0.0622 | 0.0783 | 19.07 | 0.0799 | 0.0617 | 69.06 | 0.0546 | 0.0276 | 18.49 | 0.0771 | 0.0386 | 26.65 | 0.1026 | 0.0650 | 92.11 |
| SSM | 0.0552 | 0.0232 | 11.82 | 0.0308 | 0.1203 | 82.99 | 0.0723 | 0.0912 | 145.27 | 0.0560 | 0.0235 | 18.27 | 0.0723 | 0.0641 | 26.09 | 0.1063 | 0.0568 | 92.15 |
| OTGNet* | 0.0648 | 0.0236 | 316.15 | 0.0868 | 0.0518 | 49.42 | 0.1031 | 0.0459 | 709.49 | — | — | — | — | — | — | — | — | — |
| URCL | 0.0562 | 0.0303 | 11.57 | 0.0726 | 0.0649 | 20.13 | 0.0915 | 0.0431 | 70.32 | 0.0584 | 0.0216 | 20.13 | 0.0902 | 0.0284 | 27.58 | 0.1089 | 0.0566 | 93.43 |
| LTF | **0.0682** | **0.0195** | 25.05 | **0.0871** | **0.0474** | 39.16 | **0.1110** | **0.0165** | 72.94 | **0.0681** | **0.0096** | 51.80 | **0.1134** | **0.0081** | 58.56 | **0.1253** | **0.0383** | 101.06 |

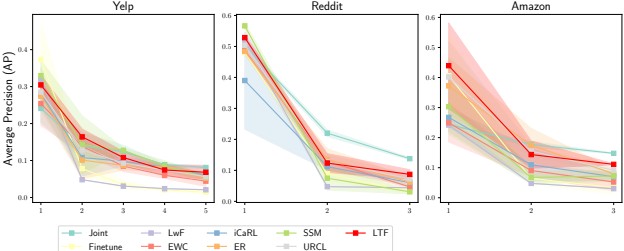

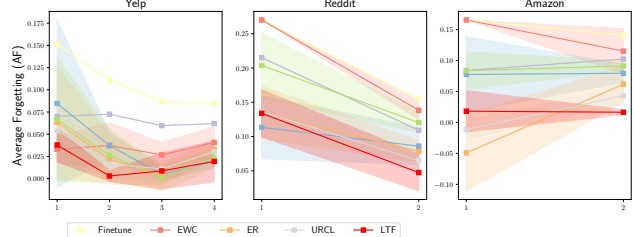

Figure 3: The average precision (AP) of LTF and the baselines at each period based on TGAT.

Figure 4: The average forgetting (AF) of LTF and the baselines at each period based on TGAT.

$N$'s update is recorded, as it accumulates the most data and is the most time-consuming.

**Implementation Details.** The experiments are run on Nvidia A30 GPU. The implementation of backbone models follow the code provided by DyGLib (Yu et al., 2023). TGAT contains two layers, and each layer has 4 attention heads. DyGFormer contains two layers, and each layer has 4 attention heads. The number of temporal neighbors is 10 and they are selected based on their freshness. We use a single 2-layer perceptron to classify all the nodes of the period, which is called class-incremental setting in continual learning. For all data sets, dropout rate is 0.4, learning rate is 0.00001, training epochs for each period is 100, and batch size is 600. Early stop is applied when validate AP does not improve for 20 epochs. For the selection based methods, 1000 events are selected for each class data at each period of Reddit and Amazon, and 500 for Yelp. Additionally, for LTF, the size of $G_N^{sim}$ is set to 500 for all data sets, and data are partitioned to have around 10000 samples in each part. The reported results are averaged over 3 random seeds. For all data sets, each period is split into 80% training, 10% validation, and 10% test. The testing data are not seen in training and validation.

### 4.2. Main Experiments

The overall comparison of LTF with other baselines is shown in Tab.2, with performance trends in Fig.3 and Fig. 4. OTGNet modifies TGAT with a unique structure, making adaptation to DyGFormer non-trivial. Naive approaches like Joint and Finetune face efficiency and effectiveness issues; Joint requires long training times, while Finetune performs worse than other baselines. Existing continual learning methods partially address TGCL, achieving higher APs (lower AFs) than Finetune with significantly lower time costs than Joint. Regularization-based methods are generally weaker than selection-based methods, highlighting the importance of data for updating old knowledge. However, a performance gap remains compared to Joint. OTGNet improves subset selection by ensuring importance and diversity, outperforming other baselines but suffering from high time complexity. LTF, with its theoretical guarantees, achieves better performance than OTGNet with lower time costs. On DyGFormer, LTF outperforms OTGNet across all datasets. Full results with standard deviations are provided in Appendix I.

### 4.3. Ablation Study

**Selection and Regularization Components.** In Tab.3, we evaluate the impact of each LTF component. The key terms of our selection objective in Eq.(8) are error $\hat{\epsilon}(\cdot|\cdot)$ and distribution $\hat{d}^2_{MMD}(\cdot, \cdot)$, which are represented by Err. and Dist. respectively. We also analyze the effect of adding $ldst(\cdot)$ to the training objective. The first two lines of Tab. 3 show that neither selection component alone is sufficient to find an effective subset. Yelp and Reddit rely more on distribution similarity, while Amazon benefits from lower error. Com-

Table 3: Ablation study on the selecting and learning components of LTF. The applied components are noted with Y. The best and second best results are noted in **Bold** and Underline.

| Component | | | TGAT | | | | | | | | | DyGFormer | | | | | | | | |
|---|---|---|---|---|---|---|---|---|---|---|---|---|---|---|---|---|---|---|---|---|
| Select | | Learn | Yelp | | | Reddit | | | Amazon | | | Yelp | | | Reddit | | | Amazon | | |
| Err. | Dist. | $l_{dst}(\cdot)$ | AP↑ | AF↓ | Time↓ | AP↑ | AF↓ | Time↓ | AP↑ | AF↓ | Time↓ | AP↑ | AF↓ | Time↓ | AP↑ | AF↓ | Time↓ | AP↑ | AF↓ | Time↓ |
| Y | | | 0.0438 | 0.0439 | | 0.0579 | 0.0736 | | 0.1063 | 0.0185 | | 0.0467 | 0.0309 | | 0.0807 | 0.0407 | | 0.1203 | 0.0456 | |
| | Y | | 0.0565 | 0.0322 | **11.79** | 0.0640 | 0.0695 | **19.28** | 0.0592 | 0.0684 | 67.77 | 0.0543 | 0.0266 | **18.51** | 0.0863 | 0.0350 | **27.12** | 0.1161 | 0.0465 | **90.18** |
| Y | Y | | 0.0654 | 0.0215 | | 0.0866 | 0.0447 | | 0.1004 | 0.0078 | | 0.0618 | 0.0155 | | 0.0939 | 0.0272 | | 0.1231 | 0.0366 | |
| Y | Y | Y | **0.0682** | 0.0195 | 25.05 | **0.0871** | 0.0474 | 39.16 | **0.1110** | 0.0165 | 72.94 | **0.0681** | 0.0096 | 51.80 | **0.1134** | 0.0081 | 58.56 | **0.1253** | 0.0383 | 101.06 |

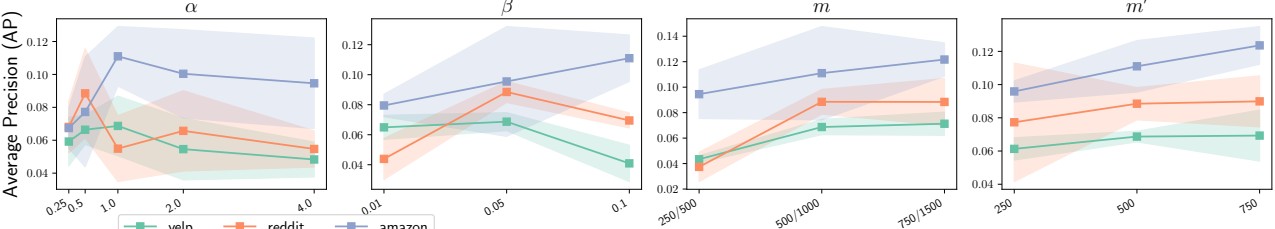

Figure 5: Sensitivity on the key hyper-parameters based on TGAT.

Table 4: Ablation study on the partition methods when selecting the subsets. The reported values are AP.

| Partition | TGAT | | | DyGFormer | | |
|---|---|---|---|---|---|---|
| Method | Yelp | Reddit | Amazon | Yelp | Reddit | Amazon |
| Kmeans | 0.0598 | 0.0754 | 0.0929 | 0.0569 | 0.0912 | 0.1104 |
| Hierarchical | 0.0613 | 0.0792 | 0.0897 | 0.0621 | 0.0987 | 0.1091 |
| Random | **0.0682** | **0.0871** | **0.1110** | **0.0681** | **0.1134** | **0.1253** |

bining both improves performance across all datasets and backbones. After selecting the most effective data, incorporating $l_{dst}(\cdot)$ further enhances performance. The additional optimization time is less significant for denser graphs, especially in Amazon, where new class data dominates. Full results are provided in Appendix J.

**Data Partition Approaches.** We reduce selection complexity by partitioning old-class data and prove that an optimal partition should preserve distribution. Tab. 4 examines different partition methods. Selecting subsets without partitioning exceeds GPU memory limits (24G on Nvidia A30), making performance untrackable. Among intuitive methods, random partitioning is more effective, as k-means or Hierarchical clustering alter data distribution of each partition, conflicting with theorem requirements. The study on the impact of partition size is included in the Appendix K.

### 4.4. Sensitivity Analysis

In Fig. 5, we evaluate the sensitivity of our method over the essential hyper-parameters on TGAT. The results on DyGFormer are in Appendix L. The empirical analysises are listed as follows, which apply to both backbones:

- **The impact of $\alpha$.** $\alpha$ balances error and distribution in selecting $G_N^{sub}$ in Eq. (8). Across values [0.25, 0.5, 1, 2, 4], Yelp and Reddit favor smaller weights, while Amazon prefers larger ones, aligning with the ablation study. Optimal performance occurs around $\alpha = 1$ for all datasets, confirming the importance of both error and distribution in subset selection.

- **The impact of $\beta$.** $\beta$ controls the weight of $l_{dst}(\cdot)$ during subset learning. Despite favoring distribution in selection, Yelp and Reddit require a low $\beta$ for regularization, while Amazon needs a higher value. This suggests that distribution is crucial for generalizing subset knowledge, with $l_{dst}(\cdot)$ enhancing its effect during learning.

- **Size $m$ of $G_N^{sub}$.** As $G_N^{sub}$ is the major carrier of the old class knowledge, its size $m$ is an important factor for the performance. Following the setup of the main experiment, the memory size is set to [250, 500, 750] for Yelp, and [500, 1000, 1500] for Reddit and Amazon. With the increase of memory size, the performance continuously improves, which is consistent with the intuition.

- **Size $m'$ of $G_N^{sim}$.** $m'$ affects the quality of distribution approximation in Eq. 10. It can be seen Amazon requires a larger size, while 500 is effective enough for Yelp and Reddit. This is because the distribution of Amazon is more complex and requires a larger sample size to approximate.

In addition to the effectiveness study, we also evaluated the efficiency-performance tradeoff by varying $m$ and $m'$ in Tab. 5. Results show that increasing either $m$ or $m'$ consistently improves average precision (AP), at the cost of longer training time. This trend is expected due to the $O(mm')$ complexity introduced by the regularization loss.

Notably, the full LTF model (with both subgraph selection and regularization) achieves the highest accuracy among all continual learning baselines, but also incurs additional runtime due to regularization. However, since our selection and regularization modules are decoupled, the regularization can be disabled when efficiency is a priority. In such cases, the selection-only variant (No Reg. for $m'$) still outperforms existing replay-based methods with comparable runtime, offering a flexible trade-off between performance and efficiency.

Table 5: Sensitivity analysis on the efficiency-performance tradeoff by varying $m$ and $m'$ on Yelp dataset and DyG-Former backbone.

| Setting | AP ↑ | Time (s) ↓ |
|---|---|---|
| *Varying $m$ (size of $G_N^{sub}$)* | | |
| $m = 250$ | 0.0434 | 34.99 |
| $m = 500$ | 0.0681 | 54.30 |
| $m = 750$ | 0.0713 | 72.31 |
| *Varying $m'$ (size of $G_N^{sim}$)* | | |
| Best Baseline | 0.0601 | 14.24 |
| $m' = 0$ (No Reg.) | 0.0618 | 18.51 |
| $m' = 250$ | 0.0624 | 36.70 |
| $m' = 500$ | 0.0681 | 54.30 |
| $m' = 750$ | 0.0693 | 71.88 |

### 4.5. Case Studies

In this sections, we include experiments on special questions of our problem, including the necessity of TGNNs in solving the proposed problem, and how more complex datasets may affect the performance of LTF.

**The necessity of TGNNs.** As node embeddings are the key media of selecting data and classifying nodes, here we study whether the embeddings along can support the whole training process, rather than using the topological structures. As shwon in Tab. 6, the MLP backbone, which only uses the node embeddings, is not able to achieve a good performance on even under the joint setting. This shows that the topological structures are essential for the TGCL problem, and the TGNNs are necessary to solve it.

Table 6: The comparison of AP across different backbone models and TGCL methods.

| | MLP | TGAT | DyGFormer |
|---|---|---|---|
| Joint | 0.0184 | 0.1477 | 0.1500 |
| Finetune | 0.0160 | 0.0340 | 0.0551 |
| LTF | 0.0171 | 0.1110 | 0.1253 |

**More Complex Datasets** To evaluate the scalability and robustness of our approach under more challenging conditions, we construct two new large-scale benchmarks, Reddit-Large for more data updates and Reddit-Long for longer period durations:

- **Reddit-Large** comprises 344,630 nodes, 4,962,297 edges, and spans 16 time periods, with 2 novel classes introduced per period, totaling 32 classes.

- **Reddit-Long** consists of 558,486 nodes, 5,323,230 edges, and spans 4 time periods, each covering 180 days. In each period, 6 new classes are introduced, resulting in a total of 24 classes evenly added over time.

We evaluate several baseline methods alongside our proposed LTF framework. As shown in Table 7, despite the increased difficulty in data updates and durations, LTF maintains competitive performance in terms of both predictive accuracy and runtime, outperforming other continual learning baselines such as Finetune and iCaRL when paired with the TGAT backbone.

Table 7: Performance comparison across Reddit-Large and Reddit-Long datasets.

| Method | Reddit-Large | | Reddit-Long | |
|---|---|---|---|---|
| | AP ↑ | Time ↓ | AP ↑ | Time ↓ |
| Joint-TGAT | 0.02042 | 107.73 | 0.0734 | 174.02 |
| Finetune-TGAT | 0.00237 | 6.37 | 0.0113 | 54.16 |
| iCaRL-TGAT | 0.00747 | 14.71 | 0.0354 | 60.03 |
| LTF-TGAT | 0.01043 | 37.21 | 0.0499 | 110.35 |

## 5. Conclusion

This paper introduces a novel challenge of updating models in temporal graphs with open-class dynamics, termed temporal graph continual learning (TGCL). Unlike existing problems, TGCL necessitates adapting to both emerging new-class data and evolving old-class data, requiring model updates to be both effective and efficient. Our proposed Learning Towards the Future (LTF) method addresses TGCL by selectively learning from representative subsets of old classes, a strategy substantiated by theoretical analysis. Experiments on real-life datasets show that LTF effectively mitigates forgetting with minimal additional cost.

## Acknowledgements

Lei Chen's work is partially supported by National Key Research and Development Program of China Grant No. 2023YFF0725100, National Science Foundation of China (NSFC) under Grant No. U22B2060, Guangdong-Hong Kong Technology Innovation Joint Funding Scheme Project No. 2024A0505040012, the Hong Kong RGC GRF Project 16213620, RIF Project R6020-19, AOE Project AoE/E-603/18, Theme-based project TRS T41-603/20R, CRF Project C2004-21G, Guangdong Province Science and Technology Plan Project 2023A0505030011, Guangzhou municipality big data intelligence key lab, 2023A03J0012, Hong Kong ITC ITF grants MHX/078/21 and PRP/004/22FX, Zhujiang scholar program 2021JC02X170, Microsoft Research Asia Collaborative Research Grant, HKUST-Webank joint research lab and 2023 HKUST Shenzhen-Hong Kong Collaborative Innovation Institute Green Sustainability Special Fund, from Shui On Xintiandi and the InnoSpace GBA. Xun Jian's work is partially supported by the Fundamental Research Funds for the Central Universities No. D5000240167.

## Impact Statement

This paper presents work whose goal is to advance the field of Machine Learning. There are many potential societal consequences of our work, none which we feel must be specifically highlighted here.

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

## A. Important Notations

The important notations used in the paper are summarized in Tab. 8 below. The relationships among $G_{N-1}$, $G_N^{old}$ and $G_N^{new}$ are illustrated in Fig. 6.

Table 8: Important Notations

| Notation | Meaning |
|---|---|
| $G \sim \mathcal{G}$ | Temporal graph $G$ that follows the distribution $\mathcal{G}$ |
| $V, E, T, Y$ | Nodes $V$, events $E$, time period $T$ and class set $Y$ of $G$ |
| $e = (u_t, v_t, t)$ | An event $e \in E$ that links nodes $u_t, v_t \in V$ at $t \in T$ |
| $v_t, \mathbf{x}_t$ | The node $v$ and its feature $x$ at time $t$ |
| $T_N, T_n$ | The latest period $N$ and a past period $n < N$ |
| $h \in \mathcal{H}$ | Model $h$ from hypothesis space $\mathcal{H}$ |
| $\epsilon(\cdot), \hat{\epsilon}(\cdot)$ | Classification error on distribution and finite set |
| $d_{\mathcal{H}\Delta\mathcal{H}}(\cdot)$ | Discrepancy between two distributions |
| $\hat{d}_{MMD}(\cdot)$ | Estimated **M**aximum **M**ean **D**iscrepancy |

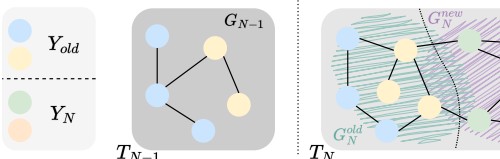

Figure 6: An illustration on the relationships among $G_{N-1}$, $G_N^{old}$ and $G_N^{new}$. $G_N^{old}$ and $G_N^{new}$ DO NOT overlap over nodes, but DO share the events connecting old and new class nodes.

## B. Additional Discussion on Related Works

Besides the related works in TGL discussed in the main content, we provide additional discussions on other TGL variants in this section. Firstly, the efficiency issue in TGL is addressed by redesigning the training framework (Zhou et al., 2022; Gao et al., 2024), sampling the representative nodes (Li & Chen, 2023), and integrating the random walk with temporal graph neural networks (Li et al., 2023).

Beyond simple temporal graphs, research has also explored temporal hyper-graphs (Yan et al., 2023), spatio-temporal graphs (Jin et al., 2023), and temporal knowledge graphs (Li et al., 2021; García-Durán et al., 2018). However, these methods typically assume a fixed set of node or entity labels and still encounter the forgetting issue when adapting to new classes, making them foundational but limited backbones for studying the TGCL problem.

There are also works addressing the OOD generalization problem in temporal graphs (Zhou et al., 2023; 2024), which address a fundamentally different problem from ours. These works aim to improve model performance on test datasets that have distributions differing from the training datasets.

In contrast, we focus on the continual learning problem that tackles the challenge of selecting new data subsets to efficiently fine-tune the past model, ensuring its effectiveness and preventing forgetting at the new period.

Some recent continual learning works have also applied the domain adaptation theory to their research, which are UDIL (Shi & Wang, 2023) and SSRM (Su et al., 2023a). However, both of them differ largely from ours. UDIL focuses on automatically finding the most suitable hyper-parameters for the losses to best balance the model stability and plasticity during learning new datasets. SSRM directly minimizes the distribution discrepancy between old and new data. Our work takes an orthogonal direction from them by selecting the most representative subset from the old class data. Besides, our TGCL problem differs from the previous continual learning settings in the considering the evolving old-class data.

## C. Proofs of Theorems

We first introduce and prove the important Lemma 3 that is originally proposed in (Ben-David et al., 2010):

**Lemma C.1.** *For any hypothesis $h$, $h' \in \mathcal{H}$ and any two different data distributions $\mathcal{D}, \mathcal{D}'$,*

$$|\epsilon(h, h'|\mathcal{D}) - \epsilon(h, h'|\mathcal{D}')| \leq \frac{1}{2}d_{\mathcal{H}\Delta\mathcal{H}}(\mathcal{D}, \mathcal{D}'), \quad (12)$$

*where $\epsilon(h, h'|\mathcal{D}) := \mathbb{E}_{x\in\mathcal{D}}[h(x) \neq h'(x)]$ is the expected prediction differences of $h$ and $h'$ on $\mathcal{D}$, and*

$$d_{\mathcal{H}\Delta\mathcal{H}}(\mathcal{D}, \mathcal{D}') := 2 \sup_{h,h'\in\mathcal{H}} |Pr_{x\in\mathcal{D}}[h(x) \neq h'(x)]$$
$$- Pr_{x\in\mathcal{D}'}[h(x) \neq h'(x)]|$$

*is the discrepancy between the two distributions.*

*Proof.* By definition, we have

$$d_{\mathcal{H}\Delta\mathcal{H}}(\mathcal{D}, \mathcal{D}') = 2 \sup_{h,h'\in\mathcal{H}} |\mathbb{P}_{x\in\mathcal{D}}[h(x) \neq h'(x)]$$
$$- \mathbb{P}_{x\in\mathcal{D}'}[h(x) \neq h'(x)]|$$
$$= 2 \sup_{h,h'\in\mathcal{H}} |\epsilon(h, h'|\mathcal{D}) - \epsilon(h, h'|\mathcal{D}')|$$
$$\geq 2|\epsilon(h, h'|\mathcal{D}) - \epsilon(h, h'|\mathcal{D}')| \quad (13)$$
$$\square$$

Based on Lemma C.1, Theorem 3.1 can be prove as follows:

**Theorem 3.1.** *Let $\mathcal{G}_N^{old}, \mathcal{G}_N^{sub}$ be the distributions of $G_N^{old}$ and $G_N^{sub}$. Let $h \in \mathcal{H}$ be a function in the hypothesis space $\mathcal{H}$ and $\tilde{h}_N^{sub}$ be the function optimized on $\mathcal{G}_N^{sub}$. The classification error on $\mathcal{G}_N^{old}$ then has the following upper bound:*

$$\min_{h\in\mathcal{H}} \epsilon(h|\mathcal{G}_N^{old}) \leq \min_{h, \mathcal{G}_N^{sub}} \epsilon(\tilde{h}_N^{sub}|\mathcal{G}_N^{old}) + \frac{1}{2}d_{\mathcal{H}\Delta\mathcal{H}}(\mathcal{G}_N^{old}, \mathcal{G}_N^{sub})$$
$$+ \epsilon(h, \tilde{h}_N^{sub}|\mathcal{G}_N^{sub}).$$

**Algorithm 1** Pseudo Code

---

1: **Input:** $G_N^{old}, G_N^{new}, \tilde{h}_{N-1}$
2: $G_N^{sub} \leftarrow \{\}, G_N^{sim} \leftarrow \{\}$
3: Partition $G_N^{old}$ into parts with sizes $p$, resulting in $W = \lceil |G_N^{old}|/p \rceil$ parts $\{G_{N,w}^{old}\}_{w=1}^{W}$
4: **for** $G_{N,w}^{old} \in \{G_{N,w}^{old}\}_{w=1}^{W}$ **do**
5:      Select $\tilde{G}_{N,w}^{sub}$ of size $m/W$ by optimizing Eq.(8)
6:      Select $\tilde{G}_{N,w}^{sim}$ of size $m/W$ by optimizing Eq.(10)
7:      $G_N^{sub} \leftarrow G_N^{sub} \cup \tilde{G}_{N,w}^{sub}, G_N^{sim} \leftarrow G_N^{sim} \cup \tilde{G}_{N,w}^{sim}$
8: **end for**
9: $\tilde{h}_N = \arg\min_{h \in \mathcal{H}} \hat{\epsilon}(h|G_N^{new}) + \hat{\epsilon}(h|G_N^{sub}) + \beta l_{dst}(G_N^{sim}, G_N^{sub}|h)$
10: **return** $\tilde{h}_N$

---

*Proof.* From the triangle inequality in (Ben-David et al., 2010) and Lemma C.1,

$$
\begin{aligned}
\epsilon(h|\mathcal{G}_N^{old}) &\leq \epsilon(h, \tilde{h}_N^{sub}|\mathcal{G}_N^{old}) + \epsilon(\tilde{h}_N^{sub}|\mathcal{G}_N^{old}) \\
&= \epsilon(h, \tilde{h}_N^{sub}|\mathcal{G}_N^{old}) + \epsilon(\tilde{h}_N^{sub}|\mathcal{G}_N^{old}) - \epsilon(h, \tilde{h}_N^{sub}|\mathcal{G}_N^{sub}) \\
&\quad + \epsilon(h, \tilde{h}_N^{sub}|\mathcal{G}_N^{sub}) \\
&\leq \epsilon(\tilde{h}_N^{sub}|\mathcal{G}_N^{old}) + |\epsilon(h, \tilde{h}_N^{sub}|\mathcal{G}_N^{old}) - \epsilon(h, \tilde{h}_N^{sub}|\mathcal{G}_N^{sub})| \\
&\quad + \epsilon(h, \tilde{h}_N^{sub}|\mathcal{G}_N^{sub}) \\
&\leq \epsilon(\tilde{h}_N^{sub}|\mathcal{G}_N^{old}) + \frac{1}{2} d_{\mathcal{H}\Delta\mathcal{H}}(\mathcal{G}_N^{old}, \mathcal{G}_N^{sub}) \\
&\quad + \epsilon(h, \tilde{h}_N^{sub}|\mathcal{G}_N^{sub}).
\end{aligned}
$$

As this inequality applies for all $h \in \mathcal{H}$ and $G_N^{sub}$, the minimum of the left side is always less than the minimum of the right side. □

## D. Pesudo Code of LTF

The pseudo code of the Learning Towards the Future (LTF) method is presented in Algorithm 1 below.

## E. Definition of RBF Kernel

**Definition** (Radial Basis Function Kernel (Schölkopf et al., 1997))**.** Consider the radial basis function kernel $\mathbb{K}$ with entries $k_{i,j} = k(x_i, x_j) = \exp(-\gamma||x_i - x_j||)$ evaluated on a sample set $X$ with non-duplicated points i.e. $x_i \neq x_j \forall x_i, x_j \in X$. The off-diagonal kernel entries $k_{i,j}, i \neq j$, monotonically decrease with respect to increasing $\gamma$.

## F. Complexity Analysis

Note that the sizes of $G_N^{old}$, $G_N^{sub}$ and $G_N^{sim}$ as $r$, $m$ and $m'$, and $G_N^{old}$ is partitioned into $W$ groups, our time complexity is analyzed as follows:

**Selection**: For each partition, we first need $O(r/W)$ to ob-

tain the errors and embeddings of all nodes. When selecting $G_N^{sub}$ from each partition, it takes $O(m/W)$ for loss estimation and $O(rm/W^2)$ for distribution estimation, which finally gives $O(m/W + rm/W^2)$. For $G_N^{sim}$, we only need $O(rm'/W^2)$ for distribution estimation. Because different partitions can be processed in parallel, the overall selection complexity is $O((r + m)/W + r(m + m')/W^2)$.

**Learning**: When learning $G_N^{sub}$, the complexity for error is $O(m)$ and that for distribution alignment is $O(mm')$. When learning $G_N^{new}$, the complexity is $O(|G_N^{new}|)$. So the overall learning complexity is $O(mm' + |G_N^{new}|)$.

## G. Data Set Details

**Yelp** Yelp is a business review dataset that contains a large amount of reviews on different businesses. When buiding Yelp data set, we regard the businesses as the nodes to construct the temporal graph. The business categories are used as the class labels. From 2015 to 2019, we take each of the five years as one period of temporal graph. The reviews from the same user within a month create events among the bussinesses they are evaluating. For each period, we select the largest three categories as the new classes and include corresponding businesses into the temporal graphs from then on. We extract word embeddings on the reviews of each business with GloVe-200d, and average these 200-dimension embeddings to get the initial node features.

**Reddit** Reddit is a online forum dataset that contains a large amount of posts and comments on different topics. When buiding Reddit data set, we regard the posts as the nodes to construct the temporal graph, following the paradigm of (Hamilton et al., 2017). The post topics are used as the class labels. We take January 1st 2017 as the start date and construct the temporal graphs of 20 days a period. The comments from the same user within 5 days create events among the posts they are commenting one. There are three periods of temporal graphs created. For each period, we select the 5 topics that have generally even number of posts as the new classes. We extract word embeddings on the comments of each post with GloVe-200d, and average these 200-dimension embeddings to get the initial node features.

**Amazon** Amazon is a product review dataset that contains a large amount of reviews on different products. When buiding Amazon data set, we regard the products as the nodes to construct the temporal graph. The product categories are used as the class labels. Starting from January 1st 2016, we take every 24 days as one period of temporal graph. The reviews from the same user within 5 days create events among the products they are reviewing. There are three periods of temporal graphs created. For each period, we select the

Table 9: The performance of different methods on the TGAT backbone. The reported values are the mean and standard deviation of AP, AF and Time.

| Method | TGAT | | | | | | | | |
|---|---|---|---|---|---|---|---|---|---|
| | Yelp | | | Reddit | | | Amazon | | |
| | AP↑ | AF↓ | Time↓ | AP↑ | AF↓ | Time↓ | AP↑ | AF↓ | Time↓ |
| Joint | 0.0810±0.0033 | — | 58.37±0.60 | 0.1378±0.0031 | — | 50.50±0.50 | 0.1477±0.0014 | — | 128.71±1.40 |
| Finetune | 0.0141±0.0045 | 0.0843±0.0000 | 9.11±0.10 | 0.0312±0.0051 | 0.1550±0.0000 | 14.93±0.14 | 0.0340±0.0213 | 0.1408±0.0000 | 65.81±2.60 |
| LwF | 0.0209±0.0050 | 0.0620±0.0028 | 13.90±0.22 | 0.0439±0.0057 | 0.1091±0.0046 | 23.67±0.44 | 0.0303±0.0097 | 0.1024±0.0105 | 102.92±3.92 |
| EWC | 0.0443±0.0142 | 0.0408±0.0178 | **9.19±0.15** | 0.0467±0.0063 | 0.1384±0.0136 | **14.95±0.14** | 0.0524±0.0292 | 0.1152±0.0363 | **68.37±3.69** |
| iCaRL | 0.0607±0.0035 | 0.0198±0.0066 | 11.57±0.14 | 0.0602±0.0076 | 0.0860±0.0274 | 19.22±0.21 | 0.0699±0.0127 | 0.0794±0.0145 | 70.03±0.48 |
| ER | 0.0521±0.0098 | 0.0332±0.0108 | 11.63±0.15 | 0.0622±0.0146 | 0.0783±0.0157 | 19.07±0.13 | 0.0799±0.0148 | 0.0617±0.0293 | 69.06±3.11 |
| SSM | 0.0552±0.0070 | 0.0232±0.0110 | 11.82±0.21 | 0.0308±0.0068 | 0.1203±0.0143 | 82.99±2.23 | 0.0723±0.0164 | 0.0912±0.0076 | 145.27±3.78 |
| OTGNet* | 0.0648±0.0120 | 0.0236±0.0139 | 316.15±17.13 | 0.0868±0.0071 | 0.0518±0.0013 | 49.42±5.55 | 0.1031±0.0259 | 0.0459±0.0232 | 709.49±42.81 |
| URCL | 0.0562±0.0091 | 0.0303±0.0085 | 11.57±0.13 | 0.0726±0.0140 | 0.0649±0.0170 | 20.13±0.55 | 0.0915±0.0065 | 0.0431±0.0130 | 70.32±2.19 |
| LTF | **0.0682±0.0108** | **0.0195±0.0130** | 25.05±0.37 | **0.0871±0.0052** | **0.0474±0.0097** | 39.16±0.62 | **0.1110±0.0018** | **0.0165±0.0041** | 72.94±2.17 |

Table 10: The performance of different methods on the DyGFormer backbone. The reported values are the mean and standard deviation of AP, AF and Time.

| Method | DyGFormer | | | | | | | | |
|---|---|---|---|---|---|---|---|---|---|
| | Yelp | | | Reddit | | | Amazon | | |
| | AP↑ | AF↓ | Time↓ | AP↑ | AF↓ | Time↓ | AP↑ | AF↓ | Time↓ |
| Joint | 0.0813±0.0038 | — | 95.11±2.04 | 0.1256±0.0140 | — | 70.64±1.22 | 0.1500±0.0054 | — | 177.38±4.27 |
| Finetune | 0.0172±0.0008 | 0.0800±0.0000 | 14.43±0.40 | 0.0360±0.0008 | 0.1433±0.0000 | 20.58±0.39 | 0.0551±0.0154 | 0.1517±0.0258 | 88.34±0.85 |
| LwF | 0.0399±0.0079 | 0.0386±0.0087 | 26.03±0.37 | 0.0469±0.0107 | 0.0944±0.0112 | 37.53±0.90 | 0.0763±0.0207 | 0.0856±0.0189 | 155.03±2.31 |
| EWC | 0.0601±0.0074 | 0.0295±0.0112 | **14.24±0.17** | 0.0521±0.0190 | 0.1046±0.0198 | **20.20±0.14** | 0.1005±0.0105 | 0.0832±0.0135 | **89.32±1.15** |
| iCaRL | 0.0558±0.0095 | 0.0214±0.0156 | 18.31±0.29 | 0.0917±0.0095 | 0.0248±0.0125 | 26.34±0.24 | 0.0945±0.0255 | 0.0775±0.0396 | 92.36±1.03 |
| ER | 0.0546±0.0066 | 0.0276±0.0094 | 18.49±0.85 | 0.0771±0.0172 | 0.0386±0.0170 | 26.65±0.30 | 0.1026±0.0029 | 0.0650±0.0057 | 92.11±0.97 |
| SSM | 0.0560±0.0009 | 0.0235±0.0044 | 18.27±0.29 | 0.0723±0.0246 | 0.0641±0.0310 | 26.09±0.59 | 0.1063±0.0197 | 0.0568±0.0170 | 92.15±1.48 |
| OTGNet* | — | — | — | — | — | — | — | — | — |
| URCL | 0.0584±0.0064 | 0.0216±0.0083 | 20.13±0.87 | 0.0902±0.0112 | 0.0284±0.0182 | 27.58±1.33 | 0.1089±0.0110 | 0.0566±0.0112 | 93.43±2.45 |
| LTF | **0.0681±0.0064** | **0.0096±0.0073** | 51.80±0.75 | **0.1134±0.0089** | **0.0081±0.0120** | 58.56±1.17 | **0.1253±0.0139** | 0.0383±0.0121 | 101.06±3.50 |

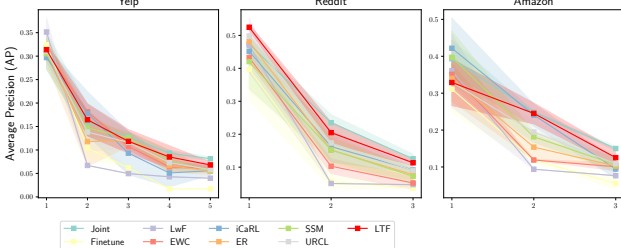

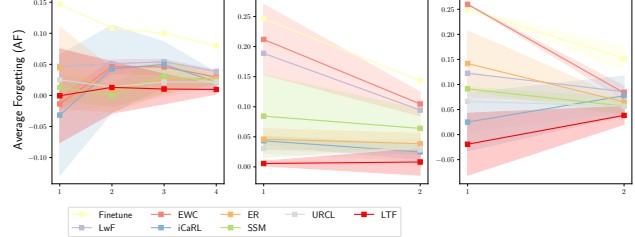

Figure 7: The average precision (AP) of LTF and the baselines at each period, based on DyGFormer.

Figure 8: The average forgetting (AF) of LTF and the baselines at each period, based on DyGFormer.

3 products that have generally even number of reviews as the new classes. We extract word embeddings on the reviews of each product with GloVe-200d, and average these 200-dimension embeddings to get the initial node features.

## H. Selection on the Hyper-parameters

The hyperparameters related to the backbone models are selected within the reported range of DyGFormer (Yu et al., 2023) based on the experience. We do not conduct further tuning on the backbone performance. The hyperparameters of our method are selected by grid search. The searching ranges and results of important hyper-parameters are reported in Sec. 4.4, with standard deviation reported as well. The hyperparameters of the baselines are selected by grid

search as well, whose names and values are: Weight of regularization loss (LwF, EWC): [0.1, 0.5, 1, 2]; Size of exemplar set (iCaRL, ER, SSM, OTGNet): 1000 for Reddit and Amazon and 500 for Yelp, which are the same as ours for fair comparison; Number of maintained neighbors (SSM): [5, 10, 20]; The other hyperparameters of OTGNet (Feng et al., 2023) are searched in the same range as reported in the original paper. The searching is performed on hyperopt package with 10 iterations.

## I. Full results on Main Experimetns

In this section, we present the results, including standard deviations, for Tab. 2 in Sec. 4.2. The detailed results for TGAT are shown in Tab. 9, and those for DyGFormer are

provided in Tab. 10. The results demonstrate that LTF not only performs well across all three datasets but also has a relatively low standard deviation, indicating the stability of the method. The standard deviation for Finetune goes to smaller than 4 digits for most datasets because its forgetting issue is sever and at the end of increments the model stably forgets most of the knowledge.

Besides, the per period performances of all methods based on DyGFormer are shown in Fig. 7 and Fig. 8. Because OTGNet is not compatible to DyGFormer, we exclude it from the presentation. The results show that LTF consistently outperforms the other methods in terms of AP and AF across all periods.

## J. Full Results on Ablation Study

The results of the ablation study in Tab. 3 with the standard deviation are shown in Tab. 12 and Tab. 13. The results demonstrate that the proposed LTF consistently outperforms the baselines across all datasets and metrics. The standard deviation of LTF is relatively low, indicating the stability of the method.

## K. Additional Study on Partition Number

In order for random partitioning to preserve the original embedding distribution, the size of each partition should be larger than a threshold. Statistically, based on the Dvoretzky–Kiefer–Wolfowitz (DKW) inequality, each partition should have 1152 samples to guarantee that the randomly sampled subset can approximate the population distribution with a 95% confidence and 0.04 approximation error. Compared with the size of our dataset (60k samples for each old class within a period in Amazon), this threshold is significantly smaller and can be easily satisfied. In our experiment, we randomly partition the dataset into parts containing 6k samples each, which is sufficient to represent the original embedding distribution.

We further study the impact of different partition sizes based on DyGFormer backbone and Amazon datasets, whose AP results are presented in Tab. 11. Performance decreases as the number of partitions increases, primarily because fewer samples in each partition result in higher selection errors. On the other hand, random constantly outperforms other clustering methods, which is consistent with our analysis that keeping the original distribution is important for effective selection.

## L. Sensitivity Analysis on DyGFormer

The additional sensitivity analysis results on DyGFormer is shown in Fig. 9. The same conclusion as in Sec. 4.4 can be drown from this set of results. This further validates the

Table 11: Comparison of Different Partition Sizes.

| Size | 10000 | 5000 | 2500 |
|---|---|---|---|
| k-Means | 0.1104 | 0.1055 | 0.0994 |
| Hierarchical | 0.1091 | 0.1063 | 0.1002 |
| Random | **0.1253** | **0.1147** | **0.1027** |

robustness of LTF on addressing TGCL.

## M. Future Directions

While this work focuses on node classification, similar challenges in integrating newly introduced, differently-distributed data are prevalent in other temporal graph tasks, such as link prediction (Di et al., 2021; Wang et al., 2021b; Di & Chen, 2023; Di et al., 2025) with new user profiles or content categories in social networks. By establishing a robust approach for handling open-class dynamics, our framework lays essential groundwork for future research. Additionally, selecting data under various scenarios (Liu et al., 2024) has also been trending recently, it is worth exploring how to extend our method to these scenarios.

Table 12: Ablation study on the selecting and learning components of LTF based on TGAT with standard deviations. The applied components are noted with Y. The best and second best results are noted in **Bold** and Underline.

| Component | | | TGAT | | | | | | | | |
|---|---|---|---|---|---|---|---|---|---|---|---|
| Select | Learn | | Yelp | | | Reddit | | | Amazon | | |
| Err. Dist. | $l_{dst}(\cdot)$ | | AP↑ | AF↓ | Time↓ | AP↑ | AF↓ | Time↓ | AP↑ | AF↓ | Time↓ |
| Y | | | 0.0438±0.0117 | 0.0439±0.0136 | | 0.0579±0.0060 | 0.0736±0.0064 | | 0.1063±0.0016 | 0.0185±0.0033 | |
| | Y | | 0.0565±0.0093 | 0.0322±0.0101 | **11.79±1.08** | 0.0640±0.0054 | 0.0695±0.0073 | **19.28±0.77** | 0.0592±0.0019 | 0.0684±0.0041 | **67.77±2.05** |
| Y | Y | | 0.0654±0.0104 | 0.0215±0.0110 | | 0.0866±0.0048 | 0.0447±0.0067 | | 0.1004±0.0023 | **0.0078±0.0032** | |
| Y | Y | Y | **0.0682±0.0108** | **0.0195±0.0130** | 25.05±0.37 | **0.0871±0.0052** | 0.0474±0.0097 | 39.16±0.62 | **0.1110±0.0018** | 0.0165±0.0041 | 72.94±2.17 |

Table 13: Ablation study on the selecting and learning components of LTF based on DyGFormer with standard deviations. The applied components are noted with Y. The best and second best results are noted in **Bold** and Underline.

| Component | | | DyGFormer | | | | | | | | |
|---|---|---|---|---|---|---|---|---|---|---|---|
| Select | Learn | | Yelp | | | Reddit | | | Amazon | | |
| Err. Dist. | $l_{dst}(\cdot)$ | | AP↑ | AF↓ | Time↓ | AP↑ | AF↓ | Time↓ | AP↑ | AF↓ | Time↓ |
| Y | | | 0.0467±0.0073 | 0.0309±0.0079 | | 0.0807±0.0084 | 0.0407±0.0220 | | 0.1203±0.0172 | 0.0456±0.0131 | |
| | Y | | 0.0543±0.0056 | 0.0266±0.0076 | **18.51±1.13** | 0.0863±0.0080 | 0.0350±0.0221 | **27.12±2.02** | 0.1161±0.0167 | 0.0465±0.0126 | **90.18±2.52** |
| Y | Y | | 0.0618±0.0060 | 0.0155±0.0082 | | 0.0939±0.0083 | 0.0272±0.0119 | | 0.1231±0.0108 | 0.0366±0.0117 | |
| Y | Y | Y | **0.0681±0.0064** | **0.0096±0.0073** | 51.80±0.75 | **0.1134±0.0089** | **0.0081±0.0120** | 58.56±1.17 | **0.1253±0.0139** | 0.0383±0.0121 | 101.06±3.50 |

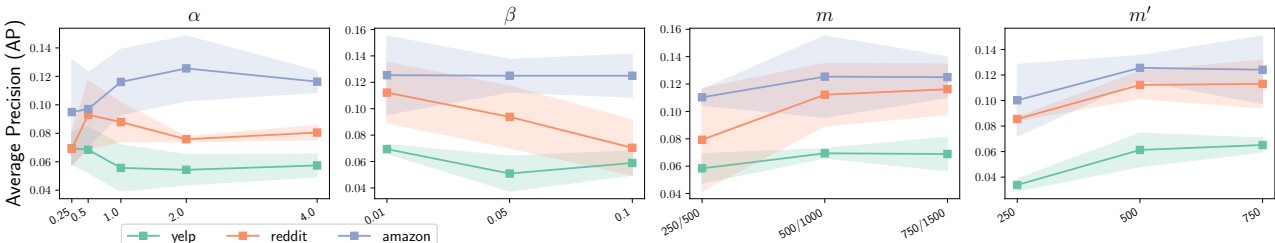

Figure 9: Sensitivity on the key hyper-parameters based on DyGFormer.

