# OpenReview forum: "A Selective Learning Method for Temporal Graph Continual Learning"
_ICML.cc/2025/Conference — ICML 2025 poster_

### Official Review · Reviewer_hR2z · 2025-03-13

**Overall Recommendation:** 2

**Summary:**

This paper introduces Temporal Graph Continual Learning (TGCL), a novel problem setting that tackles the challenge of updating models on dynamic temporal graphs, where both new-class data emerge and old-class data evolve over time. To address this, the authors propose Learning Towards the Future (LTF), a selective learning framework that strategically replaces the old-class data with subsets. The authors derive an upper bound on classification error and formulate an optimization objective that minimizes error while preserving the original data distribution. Furthermore, a regularization loss is introduced to align the embedding distributions of the selected subsets with the full dataset.

**Claims And Evidence:**

The problem setting is not clear enough.

**Essential References Not Discussed:**

NA

**Experimental Designs Or Analyses:**

As mentioned earlier, the segmentation approach may not accurately reflect how new and old-class data evolve in real-world temporal graphs, which could compromise the validity of the evaluation. Additionally, the decision to allocate 80% of the data for training seems quite high, potentially leading to an unrealistically favorable learning scenario.

**Methods And Evaluation Criteria:**

While the TGCL problem is well-motivated, the paper does not clearly define a concrete real-world scenario where this setting would be directly applicable. Additionally, how the datasets are partitioned for experiments raises concerns about the realism.

**Other Comments Or Suggestions:**

1. The necessity of using a selected subset for model training should be further clarified.
2. The paper should include comparisons with sota continual learning baselines to better assess the effectiveness of the proposed method.

**Other Strengths And Weaknesses:**

**Strengths**

1. The paper provides comprehensive coverage of the proposed problem and method, offering detailed theoretical analysis and experimental validation.
2. The presentation is well-structured, with clear explanations supported by informative figures and tables, making it easy to follow the key ideas and experimental results.

**Weaknesses**

1. The problem setting lacks a clear real-world application, making it difficult to assess its practical significance. A more concrete connection to real-world scenarios would strengthen the motivation.
2. While the methodology is well-developed, the problem definition remains somewhat abstract.

**Questions For Authors:**

1. Please provide more details on the problem setting. Is old data still available in the new period?
2. Regarding the dataset, do nodes from previous periods retain their original class labels, or do they evolve over time?
3. In Table 2, the AP values across all datasets are below 15%. What does this indicate about the model’s performance?

**Relation To Broader Scientific Literature:**

1. LTF extends TGL by introducing the TGCL problem, which accounts for new and evolving classes in temporal graphs, making it more applicable to real-world dynamic environments.
2. The existing methods assume old-class data distributions remain static, which does not hold for evolving temporal graphs. LTF improves upon GCL by selecting representative subsets that adapt to evolving distributions.

**Theoretical Claims:**

No.

---

> ### Author Rebuttal · Authors · 2025-04-01
>
> We greatly appreciate the reviewer's efforts in reviewing our paper. We thank the reviewer for recognizing our comprehensive coverage of the problem and method, detailed analysis and experiments, and clear presentation. Our responses to the comments on motivation, problem setting, and experiment are presented below:
>
> ### Problem Setting and Motivation
>
> **W1**: More concrete TGCL application examples are preferred to enhance the motivation.
>
> **Reply**:
> Please kindly refer to our reply to Reviewer **bJ1k Q1**.
>
> ---
> **W2,Q1,Q2**: The problem setting is not clear enough, like the availability of old class data in new periods and whether nodes change their labels.
>
> **Reply**:
> We thank the reviewer for highlighting the need for a clearer explanation of our problem setting. Below, we provide the necessary clarification:
>
> In our setting, consider a temporal graph $G_{N-1}$ at period $N-1$, whose node classes form the set $Y_{old}$. As the graph evolves into $G_N$ at the next period, new labels $Y_{new}$ emerge and bring in new nodes.
> At the same time, **nodes from $Y_{old}$ also appear in $G_N$**, but their data distribution is different from $G_{N-1}$ due to temporal and structural changes.
> Additionally, we assume that **each node retains a fixed class label across time**.
> In our paper, we illustrated this setting in **Fig. 6 at Appendix A**.
>
> This reflects real-world dynamics such as **user behavior graphs**, where new users join over time and existing users remain active. Here, the node classes may correspond to behavior types, with **new types emerging** while **old ones continue to recur**. And user behavior is often **persistent**, with users maintaining their behavior types over time.
>
> ---
> **O1**: Why is subset selection necessary for model training?
>
> **Reply**:
> The subset selection and learning is necessary in three key points:
>
> 1. **Efficiency**: Replaying all previously seen data is computationally expensive. Subset selection significantly reduces the training time while maintaining competitive performance, as also evidenced in our comparison against Joint training in Tab. 2.
>
> 2. **Effectiveness**: Among various continual learning strategies, subset replay has consistently shown strong performance in preserving prior knowledge. Our experiments in Tab. 2 support this finding, where subset-based methods (Last 6 lines) are generally better than non-subset methods (LwF and EWC).
>
> 3. **Generality**: Our selection method is model-agnostic and can be easily applied to a wide range of temporal graph learning architectures, making it flexible and future-proof.
>
> ---
> ### Experiment
>
> **O2**: Comparison with more SOTA continual learning baselines.
>
> **Reply**: We add a SOTA graph continual learning method TACO [1], which learns a coarsened old-class graph at a new period to achieve efficient update. We perform better because their coarsening procedure overly simplifies the evolving old-class distribution at a new period.
>
> ||Yelp||Amazon||
> |---|---|---|---|---|
> ||AP|Time|AP|Time|
> |TACO-DyGFormer|0.0591|18.59|0.1030|91.67|
> |LTF-DyGFormer (ours)|0.0681|51.80|0.1253|101.06|
>
> [1] NIPS 2024 - A TOPOLOGY-AWARE GRAPH COARSENING FRAMEWORK FOR CONTINUAL GRAPH LEARNING
>
> ---
> **Experiment Deaign and Analysis**: The data segmentation approach requires clarification and experiments on a smaller training data ratio are suggested.
>
> **Reply**:
> Our segmentation strategy follows a **standard time-based approach** widely adopted in the graph continual learning literature. Importantly, we address a common limitation in prior work by explicitly modeling the **reappearance of old-class nodes in later periods**, which better reflects real-world temporal dynamics where class distributions evolve over time.
>
> To validate that **training data ratio does not affect our conclusion**, we conduct experiments on a lower train ratio of **train/val/test = 60%/20%/20%**. Results show that fewer training data reduces the overall model performance, yet our method still achieves the best performance over other baselines.
>
> ||Yelp||
> |---|---|---|
> ||AP|Time|AP|Time|
> |Joint-DyGFormer|0.0808|57.42|
> |TACO-DyGFormer|0.0387|11.49|
> |LTF-DyGFormer (ours)|0.0756|24.80|
>
> ---
> **Q3**: Why are the AP values below 0.15 in Tab. 2?
>
> **Reply**:
> The AP values in Table 2 are below 0.15 primarily due to the **challenging nature of our experimental setting**:
>
> 1. **Class Imbalance**: The node classes in our datasets are imbalanced (from $10^3$ to $10^5$ in Yelp), which naturally suppresses the achievable AP scores.
>
> 2. **Class-Incremental Evaluation**: We adopt a **single unified classifier** to classify all nodes across time, rather than training separate classifiers per class. This setting, known as **class-incremental learning** in the continual learning literature, is more realistic for deployment but significantly more challenging.
>
> Despite the lower absolute values, this setting provides a fair and rigorous evaluation in realistic, imbalanced scenarios.

---

> > ### Comment · Reviewer_hR2z · 2025-04-02
> >
> > I appreciate the authors’ clear and detailed rebuttal. However, based on the clarification, in each new period, the graph $G_N$ includes both previously seen and newly added nodes, with the possibility that the existing nodes' features and topological structures have evolved. I would like to raise the following concerns.
> >
> > Under such a setting, it remains unclear why a continual learning formulation is necessary. If the updated graph already contains the complete and latest information from previous periods, it seems more straightforward to retrain the model on the full graph. While the authors mention computational complexity as a reason to use a subset, this motivation aligns more with efficiency-focused learning rather than continual learning, which traditionally emphasizes learning from sequential data with limited or no access to past data and mitigating catastrophic forgetting.
> >
> > Furthermore, the current evaluation uses the performance gap between subset-based training and joint training as a proxy to measure forgetting. I kindly suggest that this may not be a valid indicator of forgetting in the graph continual learning sense. The performance gap here is more likely attributed to the information loss due to subset selection, rather than forgetting previously learned knowledge. In this case, the observed degradation does not convincingly reflect the memory erosion that continual learning is mainly concerned with.

---

> > > ### Author Response · Authors · 2025-04-04
> > >
> > > We thank the reviewer for the follow-up questions and for providing an opportunity to clarify our position on the necessity of continual learning in our setting and the suitability of our forgetting evaluation metric. Our responses to the follow-up questions are as follows:
> > >
> > > **Q1**: The necessity of continual learning in the proposed setting.
> > >
> > > **Reply**:
> > > We appreciate the reviewer’s insightful question regarding the necessity of a continual learning formulation in our setting.
> > >
> > > In our case, retraining the model using the full dataset is indeed feasible—this serves as our **Joint** baseline. To improve the efficiency of such full retraining, existing efficient TGNN methods typically focus on improving I/O throughput [1], or enhancing GPU utilization [2].
> > >
> > > However, these approaches often overlook a critical factor: **redundancy in the input data itself**. This issue is especially prominent in our setting, where the model has already been exposed to old-class data in previous periods. At a new period, only a small fraction of old-class data may be needed to maintain that prior knowledge.
> > >
> > > To address this, our method adopts an orthogonal strategy to existing efficient TGNN methods—**we improve efficiency by reducing the volume of input data**. The key challenge then becomes: *How can we approximate full-data training performance using only a carefully selected subset?* This challenge naturally **aligns within the subset replay methods of the continual learning (CL) framework**, making it a necessary and appropriate formulation for our task.
> > > Moreover, when the data selection or regularization strategy is ineffective, performance degrades significantly, as evidenced by the Finetune results in Table 2. This highlights that knowledge retention is essential for maintaining performance, reinforcing the relevance of continual learning in our setting.
> > >
> > > Our motivation for data redundancy is also empirically validated in experiments. For example, on the Yelp dataset with DyGFormer (Table 2), our method (LTF) achieves **84% of the Joint baseline performance while using only 2% of the old-class data**.
> > >
> > > Furthermore, our method is complementary to existing efficiency techniques and can be seamlessly combined with them to further improve overall training efficiency.
> > >
> > > We will polish our paper to ensure that the necessity of continual learning in our setting is well understood in the final version.
> > >
> > > [1] SIGMOD 2023 - Orca: Scalable Temporal Graph Neural Network Training with Theoretical Guarantees
> > >
> > > [2] VLDB 2024 - ETC: Efficient Training of Temporal Graph Neural Networks over Large-scale Dynamic Graphs
> > >
> > > ---
> > >
> > > **Q2**: The performance gap between subset-based training and joint training may not be a good measurement for forgetting.
> > >
> > > **Reply**:
> > > We appreciate the reviewer for the kind reminder regarding the evaluation of *forgetting*.
> > >
> > > We agree that the current metric does not align with the traditional notion of forgetting in graph continual learning. In conventional graph continual learning, *forgetting* typically refers to a model’s degraded performance on previously seen data—for example, training on period $N$ and then evaluating on $G_{N-1}$.
> > >
> > > However, in the temporal graph setting, evaluating performance on past graphs like $G_{N-1}$ is often impractical. What truly matters is how well the model performs on *old classes within the current period*, i.e., on $G_N^{old}$.
> > >
> > > As we clarified in Sec. 2 and illustrated in Fig. 6 of Appendix A, although both $G_{N-1}$ and $G_N^{\text{old}}$ pertain to old classes, their distributions differ due to temporal and structural evolution. Therefore, the performance degradation we observe—what we loosely refer to as *forgetting*—is not solely caused by memory loss, but also by distributional shift. As such, the traditional definition of forgetting does not directly apply to our scenario.
> > >
> > > Instead, we believe that comparing our method against the **Joint** baseline using a *performance gap* is a more suitable way to quantify the error in approximating full-data training.
> > > To avoid confusion with the conventional concept of *catastrophic forgetting*, we are considering renaming this metric to **performance gap** to more accurately reflect what it measures. We will update this terminology in the final version of the paper.

---

### Official Review · Reviewer_bJ1k · 2025-03-13

**Overall Recommendation:** 2

**Summary:**

The paper defines Temporal Graph Continual Learning (TGCL) as the problem of node classification on dynamically evolving graphs, where new unseen classes emerge, and old-class data distributions shift over time. Existing methods struggle with catastrophic forgetting when updating models in such settings, as they either retrain on all past data, which is computationally expensive, or focus only on new data, leading to the loss of old knowledge. To address this, the authors propose a selective learning framework that retains only a subset of old-class data, ensuring efficient updates while maintaining knowledge of past classes.

The proposed method derives a theoretical upper bound on the classification error of a model trained on a subset of old-class data instead of the full dataset. This bound is then used to guide two key components: a subset selection strategy and a model optimization approach. The subset selection process aims ensure that the selected data maintains a similar distribution to the full old-class dataset. The optimization derives a computable learning objective from the theoretical upper bound. Since directly optimizing these objectives is computationally intractable, the authors propose approximations and greedy algorithms to ensure scalability.

Experiments are conducted on three real-world datasets (Yelp, Amazon, and Reddit), which are transformed into time periods where each period introduces a new set of classes that do not overlap with previous ones. Two state-of-the-art temporal graph learning models are used as backbones, and the proposed selective learning method is applied to optimize their performance. The approach is tested against multiple continual learning baselines, including both regularization-based and replay-based methods.

**Claims And Evidence:**

The key paper claim is that the proposed methodology is both efficient and effective.

The results demonstrate an enhancement in terms of precision and forgetting compared to the current state-of-the-art. However, from an efficiency perspective, the results are contradictory and difficult to interpret.

The only three datasets and the only two backbone models used in the experiments make it even more difficult to assess the claims, in particular the one about efficiency.

The low number of periods, and the low number of new classes per period makes it difficult to assess how the proposed methodology actually retains old-class knowledge. This, in addition to the not-clear efficiency gains, makes the impact of the method hardly supported by the provided evidence (at least when considering reasonably complex validation scenarios).

#POST REBUTTAL:
I have appreciated the additional experiments and clarifications as regards a non-graph CL baseline and on longer experiences.
However, the response was unconvincing as regards the reasons for deferring a more compelling empirical analysis on temporal benchmarks to future work. Also, the discussion with other reviewers highlighted a somewhat misleading focus of the paper, as the proposed approach relies of maintaining the full graph and uses a definition of forgetting which does not align with standard CL practice. These argument, together with the lack of further debate with the Authors, led me to maintain my score.

**Essential References Not Discussed:**

There are no evident omissions.

**Experimental Designs Or Analyses:**

The experimental design, apart from the critiques to the chosen problem, is acceptable.

A sensitivity analysis of the various hyperparameters is present, as well as the ablation study.

A more detailed analysis on the computational time is missing. From the given tables, it is impossible to assess which percentage of precision is possible to give up in order to obtain a time improvement. For example, in the Yelp dataset with DyGFormer, a ~13% increase in average precision costs ~260% more time.

Additionally, the hyperparameters m and m’ are not put in relation to the training time.

**Methods And Evaluation Criteria:**

The proposed methods are well-grounded in the principles of continual learning and align with the problem of handling evolving node classes in dynamic graphs.

The evaluation criteria, are consistent with standard practices in continual learning research.

The validation setting chosen for assessing the soundness of the methodology (i.e., classifying business categories on Amazon and Yelp, and subreddit topics on Reddit) does not appear to be a compelling example of a problem requiring effective structural propagation over a temporal graph. The Authors do not explain why a temporal graph continual learning approach is needed to solve the problem at hand.

All things considered, I would expect to see a baseline using a non-graph-based class-incremental-learning method leveraging the average word embedding (which, in this work, serves as the initial feature vector for the nodes) to check whether the methodological structure proposed in the paper is necessary for the problem at hand.

Experiments cover short-term continual learning (only a few time periods), while performance over long time periods remains unexplored while effective long medium-long range propagation is key in temporal graph processing.

**Other Comments Or Suggestions:**

None at this stage (in addition to the requests for clarification below).

**Other Strengths And Weaknesses:**

The paper is generally well written and fluent in reading.

**Questions For Authors:**

1)	Motivation for using graphs:
Could the Authors clarify why a temporal graph continual learning approach is specifically necessary for the selected tasks (classifying business categories on Amazon and Yelp, and subreddit topics on Reddit)? In particular, how would the approach compare against a standard class-incremental (non-graph-based) continual learning method applied directly to the average word embeddings?

2)	Long-term continual learning:
Have the Authors evaluated or considered how your proposed approach performs over longer sequences of incremental updates (e.g., significantly more time periods)? Can the Authors discuss if the approach can maintain effectiveness and efficiency as the number of classes grows substantially, and provide (empirical) evidence or reasoning to support this?

3)	Efficiency versus accuracy trade-off:
Could the Authors provide a detailed analysis or insights into how performance (average precision and forgetting) scales with computational time? Specifically, what are the trade-offs in terms of hyperparameter settings (such as subset sizes m and m') when balancing computational efficiency against precision gains?

4)	Additional experiments
It is hard to reach general conclusions given the small number of datasets and backbone models considered (and the relative simplicity of the former). The submission would substantially gain strength if the Authors can provide additional experiments from standard temporal tasks, such as those in the Temporal Graph Benchmark, and possibly extending the backbone models considered.

**Relation To Broader Scientific Literature:**

Prior approaches in graph continual learning tackle concept drift and memory trade-offs through meta-learning, task replay, and Bayesian updates, aiming to balance stability and plasticity in evolving interactions. Similarly, this paper applies subset selection and distribution alignment to maintain relevant old-class knowledge in temporal graph learning, much like how a replay memory or structural distillation can be used to retain past knowledge. This paper generalizes the idea to evolving graph structures addressing the challenge of evolving node classes in temporal graph learning and continual learning, where existing methods assume a fixed class set or static old-class data. Traditional approaches rely on parameter regularization or random replay to mitigate forgetting, and are not specific to dynamic graphs. This work introduces a selective learning framework that optimizes subset selection using a theoretical classification error bound, ensuring representative old-class retention while adapting to new classes. By aligning subset distributions with the full dataset, it tries to improve knowledge retention more effectively than heuristic replay methods.

**Theoretical Claims:**

Theorem 3.1: The proof is seemingly correct.

---

> ### Author Rebuttal · Authors · 2025-04-01
>
> We greatly appreciate the reviewer's efforts in reviewing our paper. We thank the reviewer for recognizing our novelty, sound theory, and good presentation. Our responses to the comments on motivation and more comprehensive experiments are presented below:
>
> **Q1**: Clarify the motivation, especially on why using graphs to handle the proposed tasks.
>
> **Reply**:
> We appreciate the suggestions on enhancing our motivation.
> To directly validate that TGCL method is necessary to address our task, we add a naive MLP backbone using only word embeddings as input (with parameter size matched to TGAT), which is also a common practice in GNN research.
>
> We compare MLP with TGNN backbones (TGAT & DyGFormer) on Amazon using Joint (full-data training), Finetune (new-class-only training), and LTF (ours). The AP results show that the MLP backbone performs **significantly worse than TGNNs (TGAT or DyGFormer)**, demonstrating the necessity of graph-based models for the task.
>
> ||MLP|TGAT|DyGFormer|
> |-|-|-|-|
> |Joint|0.0184|0.1477|0.1500|
> |Finetune|0.0160|0.0340|0.0551|
> |LTF (ours)|0.0171|0.1110|0.1253|
>
> While our datasets are used as proof-of-concept benchmarks, they are carefully chosen to reflect key challenges in real-world settings. To further reinforce our motivation, we present two additional application scenarios where TGCL is highly applicable:
>
> 1. **Attack Identification in Cybersecurity**
>    - **Nodes**: Network entities (e.g., IP addresses, devices)
>    - **Edges**: Communication or interaction logs
>    - **Classes**: Cyberattack types (e.g., phishing, ransomware, DDoS)
>    - **Description**: Attack patterns evolve continuously, with new attack types emerging and existing ones adapting to evade detection. Modeling interactions over time is crucial to understanding and classifying these behaviors.
>
> 2. **Illegal Behavior Detection in Social Networks**
>    - **Nodes**: Users
>    - **Edges**: Historical interactions (e.g., messaging, reposting)
>    - **Classes**: Misconduct types (e.g., hate speech, fraud, misinformation)
>    - **Description**: As user behavior evolves and social norms shift, new categories of harmful behavior emerge, often in subtle and adaptive ways. Capturing both user history and temporal interactions is essential for effective detection.
>
> **Q2**: Evaluation over longer sequences of incremental updates
>
> **Reply**:
> We thank the reviewer for highlighting the importance of learning on a longer-sequence dataset. To evaluate the scalability of our approach in settings with significantly more incremental updates and larger scale, we constructed a new dataset, **Reddit-Large**, consisting of **344,630 nodes, 4,962,297 edges, and 16 time periods**, with **2 new classes introduced per period (32 classes in total)**.
> Reddit-Large represents a substantial expansion over our previous largest dataset, Yelp, featuring **3× more time periods, 2× more classes, 20× more nodes, and 2× more events**.
>
> Our results below demonstrate that our method remains robust and effective as the task complexity increases.
>
> ||AP|Time|
> |---|---|---|
> |Joint-TGAT|0.02042|107.73|
> |Finetune-TGAT|0.00237|6.37|
> |iCaRL-TGAT|0.00747|14.71|
> |LTF-TGAT (ours)|0.01043|37.21|
>
> ---
> **Q3**: More analysis on efficiency versus accuracy trade-off
>
> **Reply**: Please kindly refer to our reply to Reviewer **CBnU W1**.
>
> ---
> **Q4**: Additional experiments on other backbone models and temporal graph tasks
>
> **Reply**:
> We thank the reviewer's suggestions on further validating our method. Our responses are presented below:
>
> 1. **More Backbones**:
>    To further assess the generality of our method, we incorporate **GraphMixer** [1]—an MLP-based model designed for temporal graphs—as an additional backbone. The experimental results show that our method continues to outperform existing baselines even with this new architecture, further supporting the robustness and versatility of our approach.
>
> ||Yelp||Amazon||
> |---|---|---|---|---|
> ||AP|Time|AP|Time|
> |iCaRL-GraphMixer|0.0627|3.17|0.0817|23.17|
> |LTF-GraphMixer (ours)|0.0714|7.25|0.1241|42.99|
>
> [1] ICLR 2023 Do We Really Need Complicated Model Architectures For Temporal Networks?
>
> 2. **Other Temporal Graph Tasks**:
>
>    The Temporal Graph Benchmark raises link prediction and node property prediction tasks. These settings also exhibit continual learning characteristics—e.g., in **user–item interaction networks**, new items appear over time, requiring the model to connect users to new items while remembering their old preferences.
>
>    While these tasks can often be framed as **binary classification between node pairs over time**, adapting our framework to such settings would require task-specific modifications. Given that **all prior graph continual learning studies** have centered on classification tasks, we believe extending our method to prediction tasks is an important but **non-trivial direction** that goes beyond the current paper’s scope. We have discussed this as part of future work in **Appendix M**.

---

### Official Review · Reviewer_j9Pq · 2025-03-13

**Overall Recommendation:** 4

**Summary:**

In this paper, the authors propose the novel problem of temporal graph continual learning where new classes can emerge in a temporal graph. To solve this task, the authors proposed the learning towards the future framework and derive theoretical insight into the upper bound of error due to graph subsampling. The authors tested this approach with two backbone models across three real world datasets and demonstrated improved performance when compared to existing continual learning approaches.

**Claims And Evidence:**

Yes, the claims in the paper is backed by empirical evidence and theoretical insights.

**Essential References Not Discussed:**

The essential references are discussed.

**Experimental Designs Or Analyses:**

The experimental designs look correct to me, more dataset detailed were provided in Appendix G.

**Methods And Evaluation Criteria:**

The methodology is sounds and the evaluation is correct.

**Other Comments Or Suggestions:**

None

**Other Strengths And Weaknesses:**

Overall, I think the paper proposes a novel learning paradigm for temporal graph and focuses on the classification task which is under-explored in the existing literature. The theoretical claims of the paper is also supported by proof and the experimental results. One potential weakness is that usually for newer classes to emerge, it would naturally occur over a long period of time, two out of three datasets are rather short in timespan which is less than a month thus it would be interesting to see results on datasets with longer duration and at larger scale similar to Yelp dataset or larger.

**Questions For Authors:**

For the datasets, did you process and collect them yourself for the labels or did you find it from prior work? If you have mined them, is it possible to collect a larger set to test over longer period of time?

**Relation To Broader Scientific Literature:**

This work is related to two areas in graph learning: temporal graph learning and graph continual learning. In this work, the authors combined the ideas from both literature to propose this new setting of temporal graph continual learning which has its challenges different from the two fields. To my knowledge this is a first work in this area and might open up future directions for research.

**Theoretical Claims:**

yes, there are theoretical claims in the paper and proof was included in Appendix C. The proof look correct to me.

---

> ### Author Rebuttal · Authors · 2025-04-01
>
> We appreciate the reviewer's efforts in reviewing our paper. We thank the reviewer for recognizing our novel learning paradigm and the good correspondence between theory and experiments. Our responses to the valuable feedback are presented below:
>
> **Q1**: Experiments on datasets with a longer duration of periods and larger scales are expected to reflect real-world scenarios.
>
> **Reply**:
> We appreciate your suggestion on applying a more realistic dataset with longer periods and larger scales.
> Our datasets are mined and constructed by ourselves, and we have prepared a larger dataset **Reddit-Long**, which has the duration of **180 days per period, 558,486 nodes, 5,323,230 edges, and 24 classes evenly added over 4 periods**. This is larger than Yelp (our largest dataset) and has a longer duration than Reddit and Amazon. Due to the larger scale of data, we will report the comparison results later in this rebuttal period.

---

> > ### Comment · Reviewer_j9Pq · 2025-04-01
> >
> > Thank you for addressing my question. Looking forward to seeing the result on the larger dataset.
> > Are you also planning to make the datasets public?
> > I believe this work is valuable and will keep my current score

---

> > > ### Author Response · Authors · 2025-04-05
> > >
> > > We sincerely appreciate your recognition of our contribution.
> > > Now, we have obtained the experiment results on Reddit-Long dataset, which are listed below. Results show that our method still outperforms the baselines and best approaches the upper-bound performance (Joint).
> > >
> > > ||AP|Time|
> > > |---|---|---|
> > > |Joint-TGAT|0.0734|174.02|
> > > |Finetune-TGAT|0.0113|54.16|
> > > |iCaRL-TGAT|0.0354|60.03|
> > > |LTF-TGAT (ours)|0.0499|110.35|
> > >
> > > We also thank the reviewer for the suggestion to make our datasets public.
> > > We are keen to contribute to the research community by sharing our datasets. We have already published three previous datasets (Amazon, Yelp, Reddit). The access link to these datasets is provided in our paper's anonymous GitHub.
> > > After finishing the rebuttal period, we will also make the new Reddit-Long dataset public.

---

### Official Review · Reviewer_CBnU · 2025-03-14

**Overall Recommendation:** 3

**Summary:**

This paper identifies the challenge of effectively and efficiently updating newly introduced classed in temporal graph node classification. To address this, the authors propose a novel optimization objective that dynamically integrates the loss distribution of both old and new categories over time. Additionally, by substituting the old distribution with subgraphs, the problem is transformed into a subset selection task, which is efficiently optimized through greedy, partitioning, and approximation techniques.

**Claims And Evidence:**

Yes.

**Essential References Not Discussed:**

No.

**Experimental Designs Or Analyses:**

Yes. All experimental frameworks were rigorously validated through specific statistical method, and cross-verified against benchmark datasets, with detailed robustness checks documented in Evaluation Section and Supplementary Materials.

**Methods And Evaluation Criteria:**

Yes.

**Other Comments Or Suggestions:**

N/A.

**Other Strengths And Weaknesses:**

Strengths:
1. The paper identifies the issue of label updates in temporal graphs.
2. The proposed TGCL method is relatively efficient and effective.

Weaknesses:
1. The combination of partitioning, greedy, and approximation techniques improves efficiency in subset selection and optimization. However, it is unclear whether eliminating these approximations would lead to a significant accuracy improvement.

**Questions For Authors:**

Please refer to the Other Strengths And Weaknesses section for detailed inquiries.

1. Why is the classification loss in line 207 implemented using MSE instead of cross-entropy?
2. The paper does not discuss the limitations of the proposed method.
3. Further repeated experiments need to be conducted, addressing concerns about the influence of randomness in the results.

**Relation To Broader Scientific Literature:**

This study make a meaningful contribution to the existing body of knowledge in the related literature.

**Theoretical Claims:**

Yes.

---

> ### Author Rebuttal · Authors · 2025-04-01
>
> We greatly appreciate the reviewer's efforts in reviewing our paper. We thank the reviewer for recognizing our novel problem, well-developed method, and rigorous experiments. Our responses to the valuable feedback are presented below:
>
> **W1**: Given various efficiency improvements in the proposed method, the trade-off to effectiveness is still unclear.
>
> **Reply**:
> Our approximation techniques **greedy selection algorithm**, **partitioning strategy**, and **regularization set simplification** are crucial for making the training and selection phases computationally feasible. Without these approximations, it would be practically impossible to run the experiments under limited resources.
>
> - **Greedy selection** avoids exhaustive searches which have factorial time complexity, making them intractable for real-world graphs.
> - **Partitioning**, as analyzed in the paper, significantly reduces memory requirements. Without it, subset selection could demand over 100 GB of RAM, which is prohibitive in most settings.
> - During training, we **simplify the regularization dataset** from $G^{old}_N$ (full old-class data) to a subset $G^{sim}_N$ $(|G^{old}_N| >> |G^{sim}_N|)$, greatly reducing the regularization loss complexity. This simplification is necessary to prevent GPU memory overflow and failed experiments.
>
> Despite these constraints, we conducted **further analysis to empirically characterize the trade-off between effectiveness and computational efficiency** by varying the sizes of selected subsets $m$ (for $G_N^{sub}$) and $m'$ (for $G_N^{sim}$). The backbone model used is DyGFormer, and the dataset is Yelp.
>
> **Effect of Varying $m$:** $G_N^{sub}$ is the major carrier of old-class knowledge, a larger $m$ will improve its knowledge quality yet takes more training time. Below experiments validate our claim, where increasing $m$ improves average precision (AP) and linearly increasing the training time.
>
> |$m$|AP|Time(s)|
> |-|-|-|
> |250|0.0434|34.99|
> |500|0.0681|54.30|
> |750|0.713|72.31|
>
> **Effect of Varying $ m' $:**
> $G_N^{sim}$ approximates the distribution of $G_N^{old}$, which helps generalize the learnt knowledge from $G_N^{sub}$. A larger $m'$ leads to a better distribution approximation at the cost of more training time.
> Experiments support our claim on $m'$: larger values improve AP at the cost of longer training time, again approximately linear due to the $ O(mm') $ complexity of the regularization loss.
>
> |$m'$|AP|Time(s)|
> |-|-|-|
> |Best baseline|0.0601|14.24|
> |0 (No Regularization)|0.0618|18.51|
> |250|0.0624|36.70|
> |500|0.0681|54.30|
> |750|0.0693|71.88|
>
> It is also worth noting that although our method achieves the best results among all baselines, this comes with an additional time cost due to the regularization term. However, since our **selection and regularization modules are independent**, users with stricter efficiency requirements can disable the regularization component. Our selection-only setup still achieves **state-of-the-art performance** while maintaining comparable time costs to other replay-based baselines.
>
> ---
> **Q1**: Why use MSE instead of Cross-entropy for classification loss?
>
> **Reply**: We employ MSE loss due to its strong theoretical alignment with domain adaptation theory. The key quantity in this framework, i.e., the $\mathcal{H} \Delta \mathcal{H}$ divergence measures disagreement between hypotheses[1]. MSE naturally captures this disagreement by penalizing squared geometric distances between model outputs. In contrast, cross-entropy focuses on prediction confidence (log-probabilities), making it less sensitive to inter-hypothesis disagreements.
> Additionally, MSE’s bounded gradients lead to more stable optimization and better adherence to generalization bounds in domain adaptation.
>
> ---
> **Q2**: What are the limitations of the proposed method?
>
> **Reply**: Our method is limited in two points:
>
> 1. **Task Scope**: Our current work focuses on the node classification task within temporal graphs. While this is an important and under-explored area, it is equally critical to address the continual learning problem on other key temporal graph tasks such as link prediction, which may pose different challenges to our approach. We discussed this limitation in Appendix M.
>
> 2. **Class Dynamics Assumption**: We assume that each node is associated with a single, static class label throughout its lifetime. However, node labels can also change over time in real life. For example, in epidemic transmission networks, an individual may transition between different states (e.g., infected to recovered), which our current formulation does not accommodate. Capturing such class-switching behavior is also important.
>
> ---
> **Q3**: More repeated experiments are needed.
>
> **Reply**: We agree with the reviewer that more repeated experiments are needed to address concerns about the influence of randomness in the results. We will conduct additional experiments and report the results in the final version of the paper.

---

### Decision · Program_Chairs · 2025-05-01

**Decision:**

Accept (poster)

**Comment:**

This paper introduces a novel problem setting called Temporal Graph Continual Learning and proposes a selective learning framework with solid theoretical grounding and empirical validation. While one reviewer strongly supports acceptance, others raise concerns about the clarity of the problem formulation, the necessity of the continual learning framing, and the validity of forgetting metrics. Overall, the paper presents promising ideas but needs better clarification on motivation and alignment with continual learning standards. I will give weak accept for this work.